Corrected: Publisher correction

# A downy mildew effector evades recognition by polymorphism of expression and subcellular localization

Shuta Asai[1,2], Oliver J. Furzer[2,4], Volkan Cevik[2,5], Dae Sung Kim[2,6], Naveed Ishaque[2,7], Sandra Goritschnig[3], Brian J. Staskawicz[3], Ken Shirasu [1] & Jonathan D.G. Jones[2]

Pathogen co-evolution with plants involves selection for evasion of host surveillance systems. The oomycete *Hyaloperonospora arabidopsidis* (*Hpa*) causes downy mildew on Arabidopsis, and race-specific interactions between Arabidopsis accessions and *Hpa* isolates fit the gene-for-gene model in which host resistance or susceptibility are determined by matching pairs of plant *Resistance* (*R*) genes and pathogen *Avirulence* (*AVR*) genes. Arabidopsis Col-0 carries *R* gene *RPP4* that confers resistance to *Hpa* isolates Emoy2 and Emwa1, but its cognate recognized effector(s) were unknown. We report here the identification of the Emoy2 *AVR* effector gene recognized by *RPP4* and show resistance-breaking isolates of *Hpa* on *RPP4*-containing Arabidopsis carry the alleles that either are not expressed, or show cytoplasmic instead of nuclear subcellular localization.

[1] Center for Sustainable Resource Science, RIKEN, 1-7-22 Suehiro-cho, Tsurumi, Yokohama, Kanagawa 230-0045, Japan. [2] The Sainsbury Laboratory, Norwich Research Park, Norwich NR4 7UH, UK. [3] Department of Plant and Microbial Biology, University of California, Berkeley, CA 94720, USA. [4] Present address: Department of Biology, University of North Carolina at Chapel Hill, Chapel Hill 27599 NC, USA. [5] Present address: Department of Biology & Biochemistry, University of Bath, Bath BA2 7AY, UK. [6] Present address: Department of Plant Sciences, College of Life Sciences, Wuhan University, Wuhan 430072, China. [7] Present address: Heidelberg Center for Personalized Oncology, DKFZ-HIPO, DKFZ, Heidelberg 69120, Germany. These authors contributed equally: Oliver J. Furzer, Volkan Cevik. Correspondence and requests for materials should be addressed to S.A. (email: shuta.asai@riken.jp) or to J. D. G.J. (email: jonathan.jones@tsl.ac.uk)

Plants and pathogens have co-evolved in a defensive and offensive battle for survival. Pathogens promote infection success by secreting effector proteins that modulate a variety of plant cellular functions, thus rendering hosts more susceptible. In turn, plants have evolved intracellular receptors containing nucleotide-binding and leucine-rich repeat domains (NLRs), that can directly or indirectly detect pathogen effectors. Disease *Resistance (R)* genes usually encode NLR proteins, and extensive genetic variation is observed at NLR-encoding loci. Recognized effectors are encoded by so-called *Avirulence (AVR)* genes and recognition by R proteins leads to effector-triggered immunity (ETI), often culminating in a hypersensitive response (HR) cell death[1]. Allelic variation, including loss-of-function mutations, in an *AVR* gene can enable a pathogen race to evade recognition and cause disease on plants that carry the cognate *R* gene.

Arabidopsis is a host for the biotrophic oomycete *Hyaloperonospora arabidopsidis* (*Hpa*; formerly *Peronospora parasitica* or *Hyaloperonospora parasitica*), a downy mildew pathogen. This model pathosystem has revealed cognate host *R* and pathogen *AVR* genes, termed *RPP* (recognition of *Peronospora parasitica*) and *ATR* (*Arabidopsis thaliana* recognized), respectively[2]. Six *RPP* loci have been cloned[3]. The corresponding recognized *Hpa* effectors have been identified for *RPP1, RPP5, RPP13,* and *RPP39,* which recognize ATR1, ATR5, ATR13, and ATR39, respectively[4–7]. Many oomycetes including *Hpa* encode secreted proteins with an RxLR (or RxLR-EER) motif that is cleaved in the pathogen upon infection[8]. Previously, we defined a total of 475 *Hpa* gene models that encode effector candidates in the reference *Hpa* isolate Emoy2[9] by applying the following criteria[10]: (1) proteins with a signal peptide and canonical RxLR motif, like ATR1, ATR13, and ATR39 (HaRxLs)[4,6,7], reported by Baxter et al.[9], (2) RxLR-like proteins with at least one non-canonical feature, like ATR5 (HaRxLLs)[5], (3) putative Crinkler-like proteins with RxLR motif (HaRxLCRNs)[11], (4) homologous proteins based on amino acid sequence similarity over the 5′ region including a signal peptide and RxLR motif (e.g., HaRxL1b, HaRxLL2b, and HaRxLCRN3b).

In Arabidopsis Col-0, *RPP4* confers resistance to *Hpa* isolates Emoy2 and Emwa1[12], but its cognate *AVR* effector gene(s) were not identified. We report here, using comparative genomics and transcriptomics among different isolates of *Hpa*, that the effector candidate *HaRxL103* corresponds to the *AVR* gene. We also show that different *Hpa* resistance-breaking strains evade detection by *RPP4* using two distinct mechanisms.

## Results

**Identification of *RPP4*-recognized effectors.** Genome sequences and expression data during infection for *Hpa* Emoy2 and Waco9 were previously reported[9,10]. Here, we sequenced genomes of five other *Hpa* isolates (Emwa1, Cala2, Emco5, Maks9, and Hind2). As reported for other filamentous plant pathogens[13], local biases were observed in the ratio of non-synonymous and synonymous nucleotide substitutions in predicted effector-encoding genes (Table 1 and Supplementary Data 1 and 2), suggesting that these genes might be under diversifying selection to evade recognition by cognate *RPP* genes. Of these seven *Hpa* isolates, Emoy2 and Emwa1 are recognized by Col-0 *RPP4*[12,14]. In *Hpa* isolates, such as Waco9, that evade *RPP4* detection, *Hpa* effector(s) recognized by *RPP4* could be deleted, polymorphic, or not expressed. We investigated such possible variation with comparative genomics and transcriptomics, using transcriptome datasets of *Hpa* Emoy2 and Waco9 during infection[10]. In *Hpa* Emoy2-infected Arabidopsis Col-0 (an incompatible interaction), transcripts from *Hpa* clearly decreased from 1 day post-inoculation (dpi), consistent with *Hpa* Emoy2 growth being arrested upon recognition by

*RPP4.* The 65 predicted *Hpa* effectors expressed at 1 dpi in *Hpa* Emoy2 are thus strong candidates for an effector recognized by *RPP4.* We also examined the genomes of seven sequenced *Hpa* isolates and analyzed transcriptome data of *Hpa* Waco9. These analyses revealed five candidate *Hpa* effectors[15,16] (Fig. 1a and Supplementary Data 3). HaRxL103 and HaRxL71 were prioritized because they were expressed at 1 dpi in *Hpa* Emoy2, but not expressed in *Hpa* Waco9, during infection. HaRxL60 and HaRxL1b were identical to alleles present in *Hpa* Emwa1, yet were found to be polymorphic in *Hpa* Waco9. HaRxLL447 was selected by identifying secreted proteins whose polymorphisms associated with recognition phenotypes among the seven sequenced *Hpa* isolates (Supplementary Fig. 1).

**HaRxL103$^{Emoy2}$ is an *Hpa* effector recognized by *RPP4*.** To identify effector(s) recognized by *RPP4*, the five selected GFP-fused candidates were transiently co-expressed with FLAG-tagged RPP4 via *Agrobacterium* in leaves of *Nicotiana benthamiana* (Supplementary Fig. 2a). GFP-HaRxL103$^{Emoy2}$, but not GFP and the other GFP-fused candidates, induced *RPP4*-dependent HR within 3 days (Fig. 1b). We confirmed that *HaRxL103* is expressed at 1 dpi in *Hpa* Emoy2, but not in *Hpa* Waco9, during infection on Arabidopsis Col-0 (Fig. 1c). To evaluate the expression patterns of *HaRxL103* in a compatible interaction, we inoculated Arabidopsis *enhanced disease susceptibility 1* mutant Ws-2 *eds1-1* with *Hpa* Emoy2 and Waco9. Ws-2 *eds1-1* is susceptible to both *Hpa* Emoy2 and Waco9[17]. *HaRxL103* was induced at 1 dpi in *Hpa* Emoy2, but not in *Hpa* Waco9, during infection on Ws-2 *eds1-1* (Fig. 1c).

*RPP4* encodes an N-terminal TIR domain-containing NLR (TIR-NLR). As *EDS1* is required for the function of TIR-NLR proteins[18], we tested if *EDS1* is required for *RPP4* function. In *N. benthamiana* leaves in which the homolog of *EDS1, NbEDS1,* was transiently silenced by overexpressing hairpin RNA of *NbEDS1* (NbEDS1-RNAi), estradiol-inducible GFP-HaRxL103$^{Emoy2}$ (Est-103$^{Emoy2}$) was co-expressed with RPP4-FLAG, and then GFP-HaRxL103$^{Emoy2}$ was induced by infiltration with estradiol. HR cell death induced by GFP-HaRxL103$^{Emoy2}$ and RPP4-FLAG co-expression was observed in a leaf area overexpressing hairpin RNA targeted against *GUS* (GUS-RNAi) as a control, whereas the HR cell death was compromised in *NbEDS1*-silenced leaf area (Fig. 2a, b).

We tested in planta interaction between GFP-HaRxL103$^{Emoy2}$ and RPP4-FLAG by co-immunoprecipitation (Fig. 2c and Supplementary Fig. 3). To test if HaRxL103$^{Emoy2}$ is recognized by *RPP4* in Arabidopsis, we created transformants containing Est-103$^{Emoy2}$ and estradiol-inducible *GFP* (Est-GFP) as a control in Arabidopsis Col-0 or Col-0 *rpp4* mutant (Fig. 2d). As no visible HR cell death was observed after treatment with estradiol in the transformants, expression of *PR1,* a defense marker gene, was investigated. Strong induction of *PR1* was observed in Col-0 Est-103$^{Emoy2}$, but not Col-0 Est-GFP, after treatment with estradiol, whereas Col-0 *rpp4* Est-103$^{Emoy2}$ showed lower *PR1* expression after the treatment compared to Col-0 Est-103$^{Emoy2}$ (Fig. 2e). We tested if HaRxL103$^{Emoy2}$ induction in Col-0 could activate disease resistance against virulent *Hpa* Waco9. No *Hpa* Waco9 sporulation was observed on Col-0 Est-103$^{Emoy2}$ pretreated with estradiol (Fig. 2f). These results indicate that HaRxL103$^{Emoy2}$ is recognized by *RPP4* in an *EDS1*-dependent manner.

**HaRxL103$^{Emoy2}$ recognition requires RPP4 nuclear localization.** To check in planta subcellular localization of HaRxL103$^{Emoy2}$, GFP-HaRxL103$^{Emoy2}$ was transiently expressed in *N. benthamiana* leaves, and fluorescence was observed. Fluorescence

**Table 1 Number of synonymous and non-synonymous polymorphisms in *Hpa* isolates**

| | All genes | | | Predicted effectors | | |
|---|---|---|---|---|---|---|
| | Synonymous | Non-synonymous | Ratio[a] | Synonymous | Non-synonymous | Ratio[a] |
| Emoy2 | 3663 | 5998 | 1.6 | 120 | 525 | 4.4 |
| Emwa1 | 7706 | 12,802 | 1.7 | 240 | 923 | 3.8 |
| Waco9 | 12,792 | 18,322 | 1.4 | 328 | 1560 | 4.8 |
| Cala2 | 10,042 | 22,703 | 2.3 | 230 | 1175 | 5.1 |
| Emco5 | 10,935 | 17,859 | 1.6 | 309 | 1386 | 4.5 |
| Maks9 | 12,512 | 21,331 | 1.7 | 360 | 1656 | 4.6 |
| Hind2 | 11,635 | 16,534 | 1.4 | 309 | 1419 | 4.6 |

[a]Ratio of non-synonymous to synonymous polymorphisms

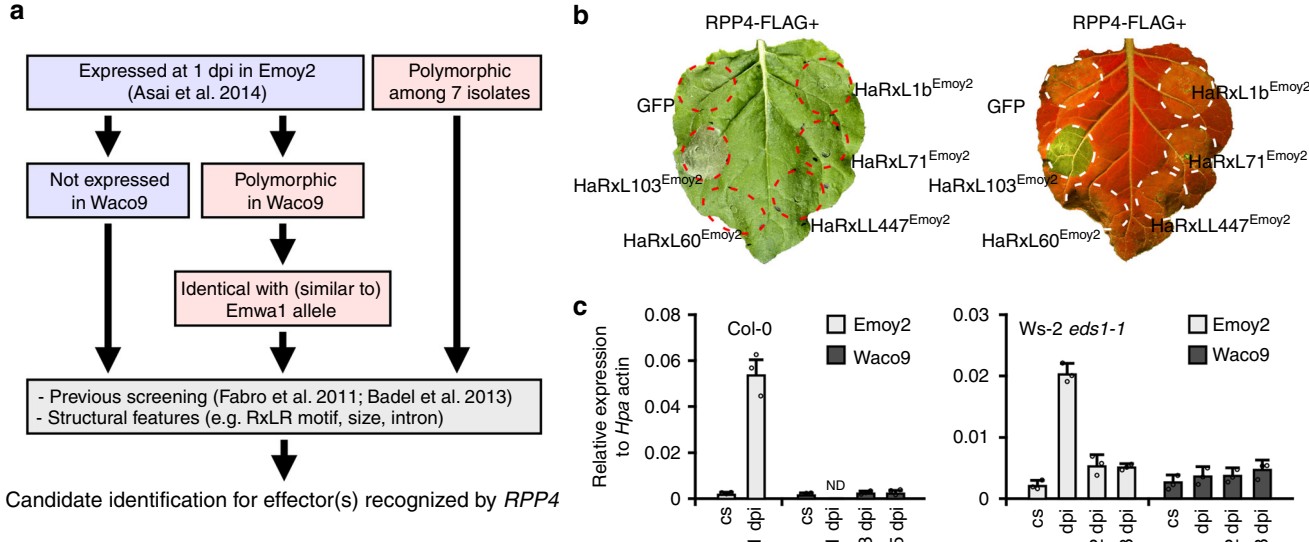

**Fig. 1** Identification of candidate effector(s) recognized by *RPP4*. **a** Flowchart to identify candidate effector(s) recognized by *RPP4*. In *Hpa* effectors expressed at 1 dpi in *Hpa* Emoy2, not-expressed effectors in *Hpa* Waco9 during infection and effectors that are polymorphic in *Hpa* Waco9 and identical with or similar to *Hpa* Emwa1 alleles were selected. Also, *Hpa* effectors in which polymorphism among 7 different *Hpa* isolates are consistent with recognition phenotypes by *RPP4* were selected. After checking results in previous screening reported by Fabro et al.[15] and Badel et al.[16] and structural features, the candidates tested were identified. **b** HR cell death phenotypes when the candidates were co-expressed with RPP4 in *N. benthamiana*. The leaves inoculated with *Agrobacterium* containing the indicated gene constructs were photographed at 3 dpi. The right one was photographed under UV to facilitate visualization of cell death. **c** Expression of *HaRxL103* in conidiospores (cs) of *Hpa* Emoy2 and Waco9 and the infections in Arabidopsis Col-0 or Ws-2 *eds1-1* mutant. The expression level was determined by qRT-PCR using specific primers for *HaRxL103*. Expression of *Hpa* actin was used to normalize the expression value in each sample. Data are means ± SDs from three biological replicates. ND not detectable

signals were seen in cytoplasm and nucleus, and especially in the nucleolus (Supplementary Fig. 4a). We evaluated which subcellular compartment HaRxL103$^{Emoy2}$ is essential for *RPP4*-mediated recognition by expressing GFP-HaRxL103$^{Emoy2}$ attached to a nuclear export signal (NES), a nuclear localization signal (NLS), or mutated nes and nls sequences. The fluorescence signals from NES-fused and NLS-fused GFP-HaRxL103$^{Emoy2}$ were detected only in cytoplasm and nucleus, respectively, whereas mutated nes and nls ones showed similar fluorescence patterns to GFP-HaRxL103$^{Emoy2}$ (Supplementary Fig. 4a). HR cell death by overexpression of GFP-NES-HaRxL103$^{Emoy2}$, but not the other constructs, was compromised compared to that induced by GFP-HaRxL103$^{Emoy2}$ when co-expressed with RPP4-FLAG in *N. benthamiana* leaves (Supplementary Fig. 4b and c). These results suggest that in planta nuclear localization of HaRxL103$^{Emoy2}$ is essential for recognition by *RPP4*.

Like HaRxL103$^{Emoy2}$, RPP4 is localized to cytoplasm and nucleus[19]. To evaluate whether nuclear localization of RPP4 is essential for recognition of HaRxL103$^{Emoy2}$, RPP4 fused to NES or nes (RPP4-NES and RPP4-nes, respectively) were constructed. As we could observe no fluorescent signals in plant cells expressing RPP4-GFP, fractionation of cytoplasmic and nuclear proteins was done to check in planta subcellular localization of RPP4-NES/nes. RPP4-FLAG and RPP4-nes-FLAG were detected in both cytoplasmic and nuclear fractions, whereas RPP4-NES-FLAG was detected only in a cytoplasmic fraction (Supplementary Fig. 5a). RPP4-nes-FLAG induced HR cell death at the same levels as RPP4-FLAG, but HR cell death mediated by RPP4-NES-FLAG was compromised when co-expressed with GFP-HaRxL103$^{Emoy2}$ in *N. benthamiana* leaves (Supplementary Fig. 5b and c). These results suggest that nuclear localization of RPP4 is important for recognition of HaRxL103$^{Emoy2}$.

**HaRxL103$^{Hind2}$ evades recognition by changing localization**. In the seven genome-sequenced *Hpa* isolates, previous genetic studies revealed that Emoy2 and Emwa1, but not five other isolates, are recognized by RPP4[12,14]. We investigated genetic diversity of *HaRxL103* alleles among the seven *Hpa* isolates. The genomic sequence data showed that there are two non-synonymous nucleotide differences in the Waco9 and Cala2 alleles (*HaRxL103$^{Waco9/Cala2}$*), one in the Emco5 and Maks9 alleles (*HaRxL103$^{Emco5/Maks9}$*), and one in the Hind2 allele (*HaRxL103$^{Hind2}$*) compared to the Emoy2 allele (*HaRxL103$^{Emoy2}$*). Emwa1 (*HaRxL103$^{Emwa1}$*) is heterozygous and carries both Emoy2 and Waco9/Cala2 alleles (Fig. 3a and Supplementary

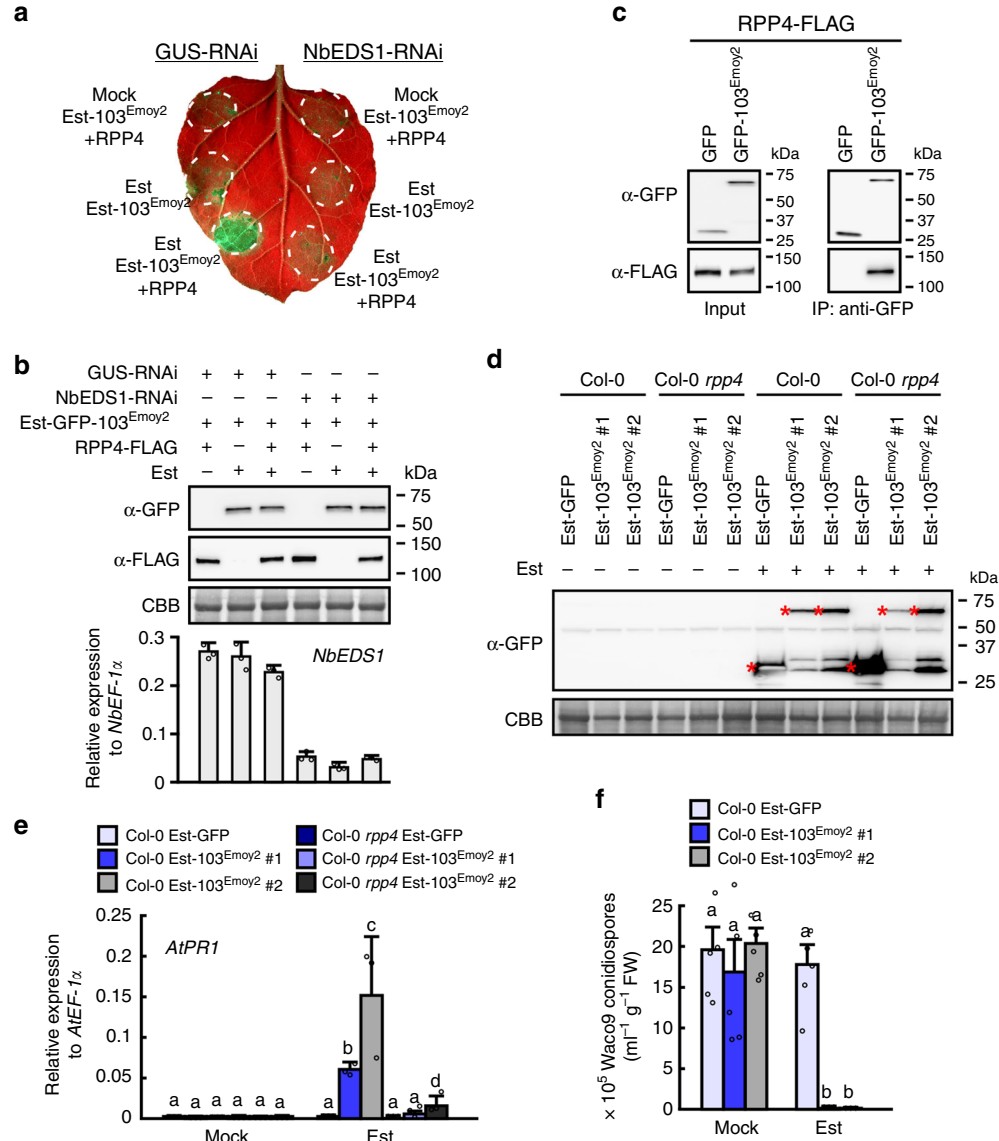

**Fig. 2** HaRxL103$^{Emoy2}$ is recognized by *RPP4* in an *EDS1*-dependent manner. **a** *N. benthamiana* leaves were inoculated with *Agrobacterium* containing estradiol-inducible GFP-HaRxL103$^{Emoy2}$ (Est-103$^{Emoy2}$), RPP4-FLAG and GUS-RNAi or NbEDS1-RNAi constructs. The different inoculation sites were infiltrated with 20 μM estradiol (Est) or water (Mock) 24 h after the inoculation. The leaves were photographed under UV at 2 days after the infiltration. **b** Confirmation of proteins accumulation and *NbEDS1* silencing. Total proteins and RNAs were prepared from *N. benthamiana* leaves described above 8 h after infiltration with estradiol or water. Immunoblot analyses were done using anti-GFP (top panel) and anti-FLAG (middle panel) antibodies. Protein loads were monitored by Coomassie Brilliant Blue (CBB) staining of the bands corresponding to ribulose-1,5-bisphosphate carboxylase (Rubisco) large subunit (bottom panel). The bar chart indicates expression levels of *NbEDS1* determined by qRT-PCR. Data are means ± SDs from three technical replicates. The experiments were repeated two times with similar results. **c** In planta interaction between HaRxL103$^{Emoy2}$ and RPP4. Co-immunoprecipitation was performed with extracts from *N. benthamiana* leaves co-expressing GFP or GFP-HaRxL103$^{Emoy2}$ with RPP4-FLAG. MACS MicroBeads with GFP antibody were used for immunoprecipitation, and anti-GFP (upper panel) and anti-FLAG (lower panel) antibodies were used to detect the related proteins in the immunoprecipitates. Protein accumulation (**d**), *AtPR1* expression (**e**), and *Hpa* growth (**f**) in Arabidopsis Col-0 and Col-0 *rpp4* transgenic lines containing estradiol-inducible GFP (Est-GFP) and Est-103$^{Emoy2}$ constructs. **d** Total proteins and RNAs were prepared from 2-week old plants 24 h after spray treatment with 40 μM estradiol or water. Immunoblot analyses were done using anti-GFP as described in (**b**). Asterisks indicate the detected GFP or GFP-HaRxL103$^{Emoy2}$ constructs. **e** The expression level of *AtPR1* was determined by qRT-PCR. Data are means ± SDs from three biological replicates. Different letters indicate significantly different values at $p < 0.05$ (one-way ANOVA, Tukey's HSD). **f** Three-week-old transgenic lines 24 h after spray treatment with estradiol or water were inoculated with *Hpa* Waco9. Conidiospores were harvested and counted at 5 dpi. Data are means ± SEs from five biological replicates. Different letters indicate significantly different values at $p < 0.01$ (one-way ANOVA, Tukey's HSD)

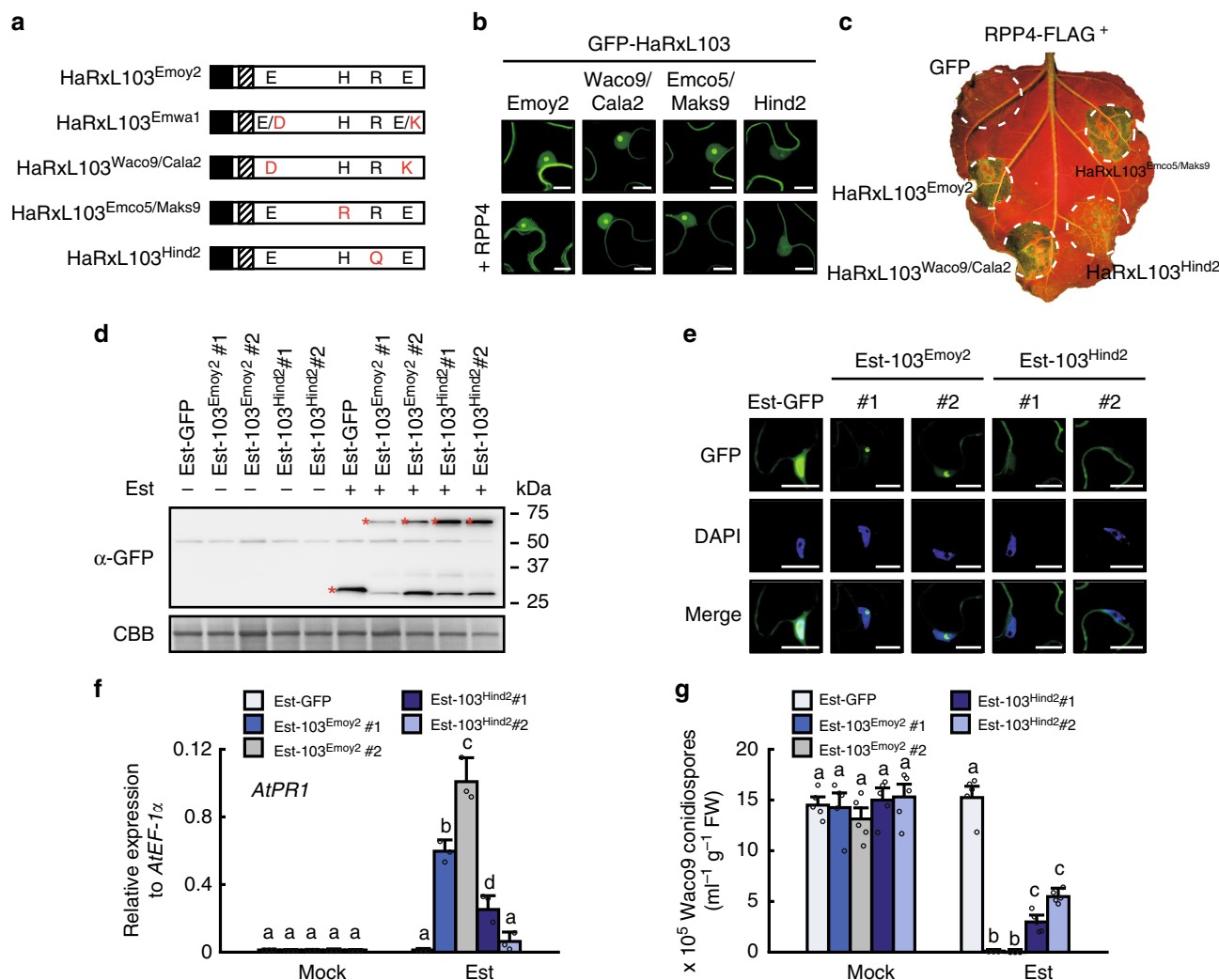

**Fig. 3** Effect of mutations in HaRxL103 alleles on in planta subcellular localization and recognition by *RPP4*. **a** Schematic structures of HaRxL103 alleles. Filled and diagonal boxes indicate an N-terminal signal peptide and an RxLR motif, respectively. Red-letter residues indicate polymorphic sites. E, Glu; H, His; R, Arg; D, Asp; K, Lys; Q, Gln. **b** Subcellular localization of HaRxL103 alleles. GFP-tagged HaRxL103 alleles were transiently (co)expressed with/without RPP4-FLAG via agroinfiltration in *N. benthamiana*. Images are from GFP channel and single-plane confocal images. Scale bars, 10 μm. **c** HR cell death phenotypes when co-expressed of HaRxL103 alleles with RPP4 in *N. benthamiana*. The leaves inoculated with *Agrobacterium* containing the indicated gene constructs were photographed under UV at 3 dpi. Protein accumulation (**d**), subcellular localization (**e**), *AtPR1* expression (**f**), and *Hpa* growth (**g**) in Arabidopsis Col-0 transgenic lines containing Est-GFP, Est-103Emoy2 and estradiol-inducible GFP-HaRxL103Hind2 (Est-103Hind2) constructs. Immunoblot, qRT-PCR and *Hpa* growth analyses were done as described in Fig. 2d–f. **d** Asterisks indicate the detected GFP, GFP-HaRxL103Emoy2 or GFP-HaRxL103Hind2 constructs. **e** Col-0 transgenic lines pretreated with estradiol were DAPI-stained. The upper image is from the GFP channel, the middle image is from the DAPI channel, and the lower image is the overlay of the GFP and DAPI channels. Scale bars, 10 μm. **f** Data are means ± SDs from three biological replicates. Different letters indicate significantly different values at $p < 0.01$ (one-way ANOVA, Tukey's HSD). **g** Data are means ± SEs from five biological replicates. Different letters indicate significantly different values at $p < 0.01$ (one-way ANOVA, Tukey's HSD)

Fig. 6). We tested in planta subcellular localization and *RPP4*-mediated HR cell death inducibility of HaRxL103 alleles. GFP-fused HaRxL103 alleles were transiently expressed with or without RPP4-FLAG in *N. benthamiana*. GFP-HaRxL103Waco9/Cala2 and GFP-HaRxL103Emco5/Maks9 were localized to cytoplasm and nucleus, especially nucleolus, as observed for GFP-HaRxL103Emoy2, whereas we observed only a weak fluorescent signal of GFP-HaRxL103Hind2 in the nucleolus (Fig. 3b; top). Their subcellular localization was unaltered by co-expression with RPP4-FLAG (Fig. 3b; bottom). GFP-HaRxL103Waco9/Cala2 and GFP-HaRxL103Emco5/Maks9 induced HR cell death indistinguishable from GFP-HaRxL103Emoy2 when transiently co-expressed with RPP4-FLAG in *N. benthamiana*, but HR cell death induced by GFP-HaRxL103Hind2 was dramatically reduced (Fig. 3c). In Col-0 transformants containing estradiol-inducible

GFP-HaRxL103Emoy2 (Est-103Emoy2) and GFP-HaRxL103Hind2 (Est-103Hind2), similar fluorescent signal patterns to ones in *N. benthamiana* transiently expressed were observed (Fig. 3b, d, and e). Consistent with HR phenotypes in *N. benthamiana*, Col-0 Est-103Hind2 showed less induction of *PR1* than Col-0 Est-103Emoy2 after treatment with estradiol (Fig. 3f). *Hpa* sporulated on Col-0 Est-103Hind2, but not on Col-0 Est-103Emoy2, pretreated with estradiol (Fig. 3g).

The finding that GFP-HaRxL103Hind2 shows little accumulation in nucleolus and less inducibility in *RPP4*-mediated HR cell death than GFP-HaRxL103Emoy2 prompted us to examine the possibility that reduced recognition of HaRxL103Hind2 by *RPP4* is due to the difference in subcellular localization. To test this hypothesis, NLS- and the mutated nls-fused GFP-HaRxL103Hind2 derivatives (GFP-NLS-HaRxL103Hind2 and GFP-nls-HaRxL103Hind2, respectively)

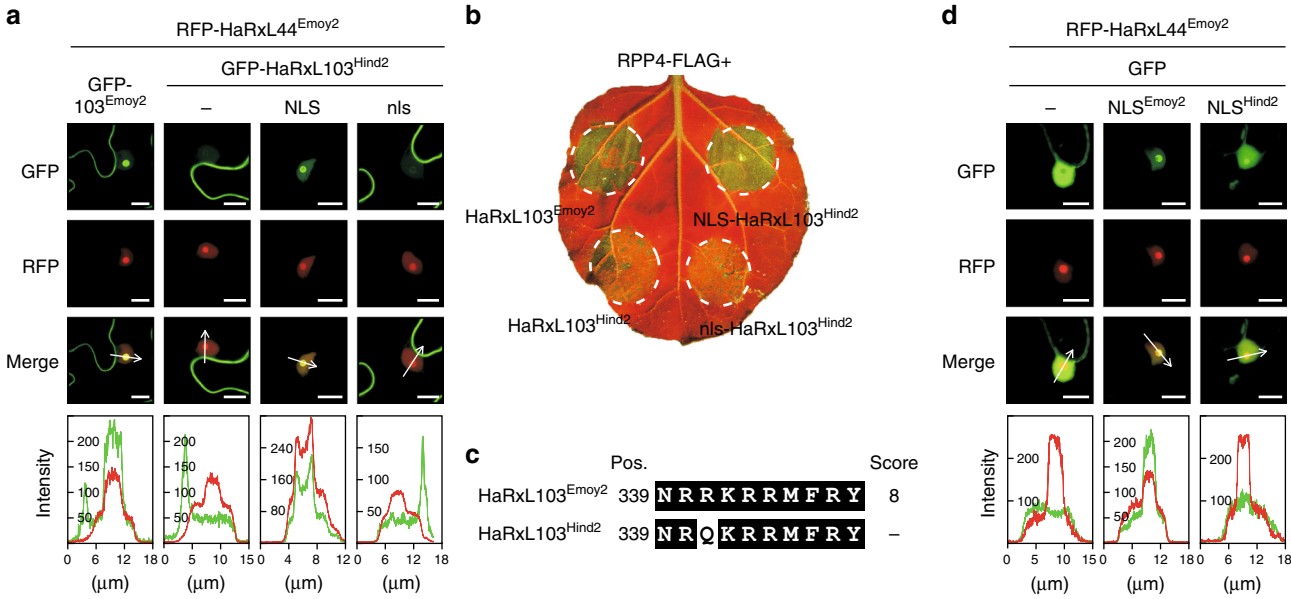

**Fig. 4** The mutation in HaRxL103[Hind2] affects in planta subcellular localization. **a** Subcellular localization of GFP-HaRxL103[Emoy2] (GFP-103[Emoy2]), GFP-HaRxL103[Hind2], and NLS- or nls-fused GFP-HaRxL103[Hind2]. GFP-tagged HaRxL103[Emoy2] and HaRxL103[Hind2] variants were transiently co-expressed with RFP-HaRxL44[Emoy2] via agroinfiltration in *N. benthamiana*. The upper image is from the GFP channel, the middle image is from the RFP channel, and the lower image is the overlay of the GFP and RFP channels. Fluorescence intensity profile (GFP, green; RFP, red) across the white arrow was performed using the analyzing software (Leica, bottom). Scale bars, 10 μm. **b** HR cell death phenotypes when co-expressed of HaRxL103[Emoy2] and HaRxL103[Hind2] variants with RPP4 in *N. benthamiana*. The leaves inoculated with *Agrobacterium* containing the indicated gene constructs were photographed under UV at 3 dpi. **c** A predicted NLS sequence from HaRxL103[Emoy2] and HaRxL103[Hind2]. Prediction of NLS in HaRxL103[Emoy2] or HaRxL103[Hind2] was done in cNLS Mapper. Position of predicted NLS and score for the prediction are indicated. Identical sequences are indicated in white on black. **d** Subcellular localization of GFP-fused predicted NLS from HaRxL103[Emoy2] (NLS[Emoy2]) and HaRxL103[Hind2] (NLS[Hind2]). GFP, GFP-NLS[Emoy2], and GFP-NLS[Hind2] were transiently co-expressed with RFP-HaRxL44[Emoy2] via agroinfiltration in *N. benthamiana*. Imaging was performed as described in (**a**). Scale bars, 10 μm

were constructed. For the subcellular localization assay, RFP-fused HaRxL44[Emoy2], an *Hpa* effector that localizes to the plant cell nucleolus[20], was co-expressed as a nucleolus marker. We observed nucleolar localization of GFP-NLS-HaRxL103[Hind2] similar to GFP-HaRxL103[Emoy2], whereas little accumulation of GFP-nls-HaRxL103[Hind2] was observed in nucleolus as for GFP-HaRxL103[Hind2] (Fig. 4a). Importantly, GFP-NLS-HaRxL103[Hind2], but not GFP-nls-HaRxL103[Hind2], induced HR cell death at the same level as GFP-HaRxL103[Emoy2] when co-expressed with RPP4-FLAG in *N. benthamiana* (Fig. 4b).

The prediction of NLS revealed that the non-synonymous single nucleotide variant (SNV) in HaRxL103[Hind2] is in a predicted mono-partite NLS, and the amino acid sequence in HaRxL103[Hind2] no longer corresponds to a predicted NLS (Fig. 4c and Supplementary Fig. 6). To evaluate if the predicted NLS is functional and if the mutation in HaRxL103[Hind2] affects the function, GFP fusions to the predicted NLS sequences from HaRxL103[Emoy2] and HaRxL103[Hind2] (GFP-NLS[Emoy2] and GFP-NLS[Hind2], respectively) were constructed. GFP-NLS[Emoy2] was visible in nucleoplasm and the nucleolus as observed for RFP-HaRxL44[Emoy2], a nucleolus-localizing *Hpa* effector[20], whereas GFP-NLS[Hind2] showed nuclear-cytoplasmic localization and little nucleolar localization compared to GFP-NLS[Emoy2] (Fig. 4d), suggesting that the predicted NLS from HaRxL103[Emoy2], but not HaRxL103[Hind2], functions as an NLS. These results indicate that the mutation in HaRxL103[Hind2] alters in planta subcellular localization, resulting in evasion of recognition by *RPP4*.

**Localization of HaRxL103-RPP4 interaction.** The requirement of HaRxL103[Emoy2] for nuclear localization to be recognized by *RPP4* (Fig. 4 and Supplementary Fig. 4) indicates the possibility that HaRxL103[Emoy2] interacts with RPP4 in nucleus. To test this

hypothesis, we performed bimolecular fluorescence complementation analysis by co-expressing nVenus/cCFP-HaRx-L103[Emoy2] with RPP4-nVenus/cCFP in all combinations, but we failed to observe fluorescent signals. Therefore, co-immunoprecipitation in a nuclear fraction was performed. Cytoplasmic and nuclear protein extracts were separately isolated from *N. benthamiana* leaves transiently expressing GFP-HaRxL103[Emoy2] and RPP4-FLAG. Co-immunoprecipitation assays in cytoplasmic and nuclear fractions revealed HaRx-L103[Emoy2]-RPP4 interaction in both cytoplasm and nucleus (Fig. 5a). We also checked whether GFP-HaRxL103[Hind2] and NLS/nls-fused GFP-HaRxL103[Hind2] interact with RPP4-FLAG in cytoplasm and/or nucleus (Fig. 5b–d). GFP-HaRxL103[Hind2] and GFP-nls-HaRxL103[Hind2] were detected only in cytoplasmic fractions, whereas GFP-NLS-HaRxL103[Hind2] was detected only in nuclear fractions. We confirmed in planta interaction of GFP-HaRxL103[Hind2] and GFP-NLS/nls-HaRxL103[Hind2] with RPP4-FLAG in each fraction, suggesting that the mutation in HaRxL103[Hind2] does not affect the interaction with RPP4.

**Virulence co-segregates with lack of *HaRxL103* expression.** *Hpa* isolates Waco9, Cala2, Emco5, and Maks9 also avoid recognition by *RPP4*, but HaRxL103[Waco9/Cala2] and HaRxL103[Emco5/Maks9] activate *RPP4*-dependent HR cell death in *N. benthamiana* (Fig. 3c). We tested if these *Hpa* isolates evade *RPP4*-mediated immunity through lack of *HaRxL103* expression, and found *HaRxL103* is not expressed in *Hpa* Waco9, but is in Emoy2, during infection (Fig. 1c).

Previously, an outcrossed F2 population of *Hpa* isolates Emoy2 and Maks9 was used to identify *Hpa* recognized effector genes for six known *R* genes including *RPP4*[21]. Positional cloning using the Emoy2/Maks9 F2 progeny identified *ATR1*, *ATR5*, and *ATR13* as

*Hpa AVR* genes recognized by *RPP1*, *RPP5*, and *RPP13*, respectively[4–6]. In the Emoy2/Maks9 F2 progeny, segregation data for the *Hpa* gene locus corresponding with recognition by *RPP4* (*ATR4*) revealed that *RPP4*-mediated immunity is controlled by a semi-dominant allele at a single locus[21]. To evaluate whether the *HaRxL103* locus is linked to *RPP4*-mediated immunity in the Emoy2/Maks9 F2 progeny, we designed a CAPS (cleaved amplified polymorphic sequence) marker based on polymorphism between Emoy2 and Maks9 *HaRxL103* alleles. The *HaRxL103* locus was unlinked to *RPP4*-dependent recognition in the Emoy2/Maks9 F2 progeny (Supplementary Table 1). Conceivably, the genetically defined *ATR4* gene regulates expression of *HaRxL103*.

We created an outcrossed progeny of *Hpa* isolates Emoy2 and Cala2, and checked phenotypes on Arabidopsis CW84, an *Hpa* susceptible recombinant inbred line generated from a cross between Col-0 and Ws-2[22], and on CW84 lines carrying transgenic *RPP4* (CW84:RPP4^Col)[12]. The Emoy2/Cala2 F1 *Hpa* showed avirulence on CW84:RPP4^Col as observed for Emoy2 (Supplementary Fig. 7), whereas the F2 progeny segregated with virulent or avirulent phenotypes on CW84:RPP4^Col. Arabidopsis Ws-2 *eds1-1* mutant[17] was susceptible to all the Emoy2/Cala2 F2 progeny. We checked the expression levels of *HaRxL103* in the individual F2 progeny by qRT-PCR at 2 and 4 dpi on Ws-2 *eds1-1*. In all F2 progeny which showed avirulence on CW84:RPP4^Col, *HaRxL103* expression was observed during at least one time point, whereas no expression was observed in virulent isolates (Fig. 6a). Genotyping using a CAPS marker designed to detect polymorphism between Emoy2 and Cala2 *HaRxL103* alleles revealed segregation of the *HaRxL103* locus in virulent isolates (Fig. 6b). These results suggest that virulence correlates with lack of *HaRxL103* expression, but does not map to the *HaRxL103* locus in an Emoy2/Cala2 F2.

## Discussion

We report here the identification of HaRxL103, an *Hpa* effector recognized by Arabidopsis *RPP4*, and two modes of evasion of recognition in different resistance-breaking strains of *Hpa*. Of seven sequenced *Hpa* isolates, Emoy2 and Emwa1 are avirulent on Arabidopsis genotypes containing functional *RPP4*, whereas Waco9, Cala2, Emco5, Maks9, and Hind2 evade recognition by *RPP4*[12,14]. Co-expression of HaRxL103^Emoy2 with RPP4 resulted in HR cell death in *N. benthamiana* in an *NbEDS1*-dependent manner, and ectopic expression of HaRxL103^Emoy2 induced

immune responses in Col-0 plants containing *RPP4*. Co-immunoprecipitation analysis revealed that HaRxL103^Emoy2 interacts with RPP4 in both cytoplasm and nucleus. We found that nuclear, and perhaps nucleolar, localization of HaRxL103^Emoy2 is required to trigger *RPP4*-mediated immune responses and that the *Hpa* isolate Hind2 evades *RPP4* recognition by a mutation in a functional NLS in HaRxL103. In contrast, *Hpa* isolates Waco9 and Cala2 evade recognition by *RPP4* through lack of *HaRxL103* expression. Finally, analyses in Emoy2/Cala2 F2 individual progeny showing virulence or avirulence on CW84:RPP4^Col revealed that virulence is associated with lack of *HaRxL103* expression, but that this lack of expression is conferred by a gene that is not linked to the *HaRxL103* locus.

During co-evolution with plants, pathogens have inactivated deleterious genes, including recognized effector genes, by diverse mechanisms such as gene loss, mutation, and gene silencing. In filamentous plant pathogens, such as *Hpa* and *Phytophthora* species, genes encoding putative effector proteins (e.g., RxLR effectors) show signatures of diversifying selection[9,13,23]. In this study, we confirmed this correlation in *Hpa* isolates (Table 1 and Supplementary Data 1 and 2). *Hpa* Waco9 evades recognition by Arabidopsis *R* gene *RPP1* through loss of its cognate recognized effector *ATR1* from its genome[10]. Virulent isolates of wheat stem rust break resistance conferred by the wheat *Sr35* resistance gene through loss of *AvrSr35* by the insertion of a mobile element[24]. ATR1 and ATR13 are extremely polymorphic and this allelic diversity enables evasion of recognition by specific alleles of their corresponding *R* genes, *RPP1* and *RPP13*[4,6,25,26]. In *Phytophthora sojae*, a key amino acid mutation in PsAvr3c impairs a physical association with the host protein GmSKRPs involved in *Rps3c*-mediated soybean immunity, resulting in evasion of recognition by *Rps3c*[27,28]. There are some non-synonymous SNVs among *HaRxL103* alleles (Fig. 3a and Supplementary Fig. 6). We identified a single point mutation in the Hind2 allele (*HaRxL103-Hind2*) that is located in a functional NLS within HaRxL103, resulting in exclusion of the protein from the nucleus (at least nucleolus), that correlates with evasion of *RPP4*-mediated immune responses (Figs. 3 and 4). This conclusion is supported by the finding that the fusion of NLS to HaRxL103^Hind2 could restore HR cell death triggered by co-expression with RPP4 (Fig. 4b). In this study, GFP-tagged proteins were ectopically overexpressed to check those subcellular localizations. Although we cannot rule out the possibility that the localization of GFP-tagged HaRxL103 does not reflect the real localization of native

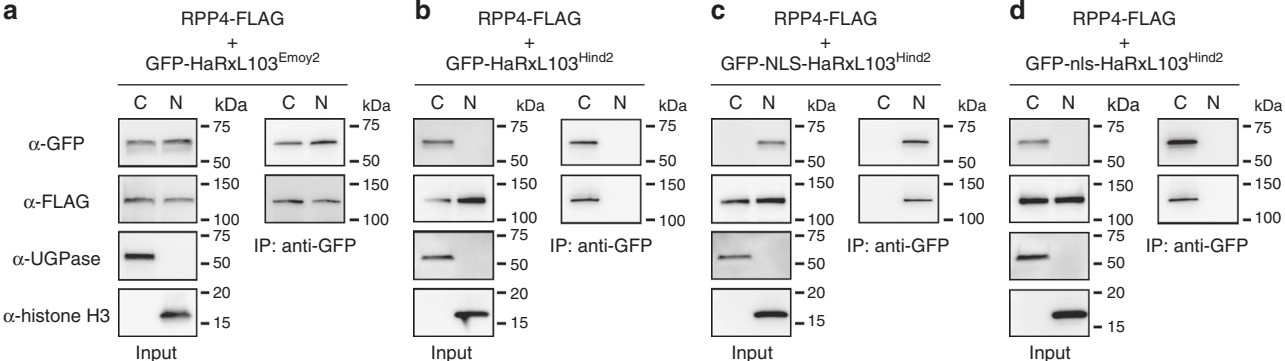

**Fig. 5** Interactions between HaRxL103^Emoy2/HaRxL103^Hind2 and RPP4 in cytoplasm and/or nucleus. In planta interaction of GFP-HaRxL103^Emoy2 (**a**), GFP-HaRxL103^Hind2 (**b**), and GFP-NLS/nls-HaRxL103^Hind2 (**c**, **d**) with RPP4-FLAG in cytoplasm and nucleus. Cytoplasmic (C) and nuclear (N) protein extracts were separately isolated from *N. benthamiana* leaves inoculated with *Agrobacterium* containing the indicated gene constructs at 2 dpi. Co-immunoprecipitation was performed with each extract using MACS MicroBeads with GFP antibody. Anti-GFP (upper panel), anti-FLAG (the 2nd panel), anti-UGPase (the 3rd panel), and anti-Histone H3 antibodies (bottom panel) were used to detect the related proteins in the immunoprecipitates. UGPase and Histone H3 were checked as markers for cytoplasmic and nuclear proteins, respectively

HaRxL103 proteins during *Hpa* infection, in planta subcellular localization of HaRxL103 correlated with *RPP4*-mediated immunity in the conditions tested. This is, to our knowledge, the first report that a pathogen effector can avoid recognition by its cognate *R* gene through changing subcellular localization in host cells.

Transcriptional silencing of *AVR* effector genes to evade resistance in host plants containing their cognate *R* genes has also been reported in *Phytophthora* species[29]. Qutob et al.[30] demonstrated transgenerational gene silencing of *P. sojae Avr3a*. Although the mechanisms responsible for its establishment remain to be determined, they found an association between small RNA accumulation and gene silencing at the *Avr3a* locus. Interestingly, comparative genomics among *Phytophthora* species revealed that RxLR effector-rich regions are enriched for genes related to epigenetic processes, suggesting a potential role for epigenetic mechanisms in oomycete pathogen evolution[23]. Despite carrying non-synonymous polymorphisms from HaRxL103$^{Emoy2}$, the HaRxL103$^{Waco9/Cala2}$ and HaRxL103$^{Emco5/Maks9}$ alleles cause *RPP4*-dependent HR cell death comparable to HaRxL103$^{Emoy2}$ in *N. benthamiana* (Fig. 3a, c). We infer that *Hpa* isolates Waco9, Cala2, Emco5, and Maks9 evade *RPP4*-mediated immunity through loss of *HaRxL103* expression. Consistent with this model, *HaRxL103* is induced during infection in *Hpa* Emoy2, but not in Waco9 (Fig. 1c), and virulence co-segregates with lack of *HaRxL103* expression in an Emoy2/Cala2 F2 (Fig. 6a). Interestingly, 2 kb upstream and 0.5 kb downstream of the *HaRxL103*-coding regions are identical between Emoy2 and Waco9, suggesting that expression of *HaRxL103* could be regulated by epigenetic mechanisms and/or specific transcriptional regulator(s) for *Hpa* isolates (discussed in the section below).

The gene-for-gene model predicts the outcome of interactions between Arabidopsis accessions and *Hpa* isolates[2]. *ATR4* was originally defined as an *Hpa* gene locus associated with the *RPP4*-mediated immunity in an Emoy2-Maks9 F2[21]. In this study, we revealed that HaRxL103 triggers *RPP4*-mediated immunity, but there is no genetic linkage between the *HaRxL103* locus and *RPP4*-mediated immunity in the Emoy2/Maks9 F2 progeny or in the Emoy2/Cala2 F2 progeny. We propose referring to HaRxL103$^{Emoy2}$ as AvrRPP4, but not ATR4. We suggest *ATR4* should be reserved for the locus at which genetic variation is found that regulates *HaRxL103* expression. The flanking sequence of the *HaRxL103*-coding region is identical in *Hpa* virulent isolate Waco9 and avirulent isolate Emoy2, so we propose that allelic variation between virulent and avirulent isolates is found in gene(s) involved in epigenetic and/or transcriptional regulation of *HaRxL103*. In flax rust, "inhibitor genes" for avirulence have also been identified[31]; this might also reflect allelic variation in a transcriptional regulator that controls expression of a recognized effector. Further analysis by positional cloning is required to uncover *ATR4*.

Effector genes that trigger ETI on their host plants are likely to be rapidly lost unless they contribute to virulence on susceptible host plants. Both ATR1 and ATR13 trigger ETI on plants that carry *RPP1* and *RPP13*, respectively, but promote virulence in a compatible interaction[32]. To evaluate whether HaRxL103$^{Emoy2}$ has a virulence function, we measured *Hpa* growth on Col-0 *rpp4* Est-GFP and Col-0 *rpp4* Est-103$^{Emoy2}$. Although *Hpa* sporulates on Col-0 *rpp4* Est-103$^{Emoy2}$ pretreated with estradiol, *Hpa* growth is reduced compared to non-estradiol-treated Col-0 *rpp4* Est-103$^{Emoy2}$ (Supplementary Fig. 8a and b), suggesting that HaRxL103$^{Emoy2}$ might still be weakly recognized in a Col-0 *rpp4* mutant. Consistent with this, although *PR1* expression is much more strongly induced in Col-0 Est-103$^{Emoy2}$ than in Col-0 *rpp4* Est-103$^{Emoy2}$, *PR1* expression is still weakly induced in Col-0 *rpp4* Est-103$^{Emoy2}$ after treatment with estradiol (Fig. 2e). *Hpa* Emoy2 sporulates on Col-0 *rpp4*, but grows better on Col-0 *eds1-*

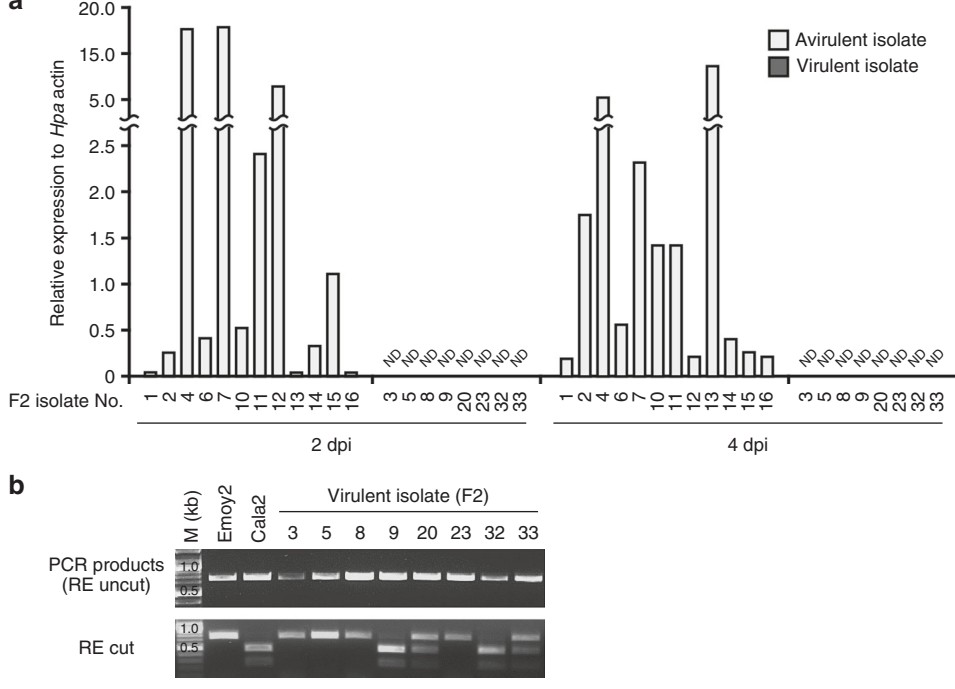

**Fig. 6** In an Emoy2/Cala2 F2, virulence consists with lack of *HaRxL103* expression. **a** Expression of *HaRxL103* during infection in Emoy2/Cala2 F2 individual progeny which showed virulence or avirulence on CW84:RPP4$^{Col}$ plants. Total RNA was prepared from the infections in Arabidopsis Ws-2 *eds1-1* mutant at 2 or 4 dpi. qRT-PCR analysis was done as described in Fig. 1c. **b** Genotyping of the *HaRxL103* locus in Emoy2/Cala2 F2 individual progeny which showed virulence on CW84:RPP4$^{Col}$ plants. Genotyping of the *HaRxL103* locus was performed by a CAPS (cleaved amplified polymorphic sequence) method. Upper and bottom panels indicate PCR products and those after restriction enzyme (RE) treatment, respectively. M marker, ND not detectable

2, whereas *Hpa* Waco9, in which *HaRxL103* is not expressed, is equally virulent on Col-0, Col-0 *rpp4*, and Col-0 *eds1-2* (Fig. 1c and Supplementary Fig. 8c). As HaRxL103[Emoy2] still induces some immune responses in Col-0 *rpp4*, we could not assess the virulence function. Loss-of-function analysis of *HaRxL103* in *Hpa* is not currently technically feasible.

If HaRxL103[Emoy2] has a virulence function, it would be interesting to investigate whether HaRxL103[Hind2] still has that function. In planta subcellular localization analysis revealed reduced accumulation of HaRxL103[Hind2] in the nucleolus compared to HaRxL103[Emoy2] (Fig. 3b). Fusion of NLS to HaRxL103[Hind2] restored accumulation in nucleolus and complemented the loss of induction of *RPP4*-mediated immunity (Fig. 4a, b). In addition, co-immunoprecipitation analysis revealed the interaction of HaRxL103[Emoy2] and HaRxL103[Hind2] with RPP4 in cytoplasm and/or nucleus (Fig. 5). Exclusion of HaRxL103 from the nucleus abolished *RPP4*-mediated HR cell death (Supplementary Fig. 3). These results suggest that the HaRxL103-RPP4 interaction in the nucleus (perhaps nucleolus) is essential for induction of *RPP4*-mediated immunity. Although the host target of HaRxL103 remains to be defined, we hypothesize that HaRxL103 exerts its virulence function in the host nucleus, and perhaps in the nucleolus. HaRxL103[Hind2] might therefore lose virulence function. The nucleolus is the site for ribosomal RNA synthesis and ribosome assembly. Recent studies revealed a role for the nucleolus in the control of various tumor suppressors and oncogenes[33]. Investigation of the HaRxL103 function might enable unknown functions of the nucleolus to be revealed.

## Methods

**Plant material and growth.** Arabidopsis plants were grown at 22 °C under a 10-h photoperiod and a 14-h dark period in environmentally controlled growth cabinets. *N. benthamiana* plants were grown at 25 °C under a 16-h photoperiod and an 8-h dark period in environmentally controlled growth cabinets. For Col-0 *rpp4* mutant, we used a homozygous line from the SALK named SALK017569[34]. A T-DNA insertion was checked by PCR using T-DNA left border primer (LBb1.3) and gene-specific primers (LP and RP) listed in Supplementary Table 2. Previously published Arabidopsis lines were: Ws-2 *eds1-1*[17], CW84[12], CW84:RPP4[Col,12], and Col-0 *eds1-2*[35].

**Pathogen assays.** *Hpa* inoculation was done as described in Asai et al.[36]. Briefly, Arabidopsis plants were spray-inoculated to saturation with a spore suspension of $1 \times 10^5$ conidiospores/ml. Plants were covered with a transparent lid to maintain high humidity (90–100%) conditions in a growth cabinet at 16 °C under a 10-h photoperiod until the day for sampling. To evaluate conidiospore production, 5 pools of 3 plants for each Arabidopsis line were harvested in 1 ml of water. After vortexing, the amount of conidiospores released was determined using a haemocytometer.

**Genome sequencing and comparative genomics of *Hpa* isolates.** Genomic DNA was extracted from *Hpa* conidiospores using a Nucleon PhytoPure DNA extraction kit (GE Healthcare) according to the procedure of the manufacturer. 36 and 76 bp paired-end libraries were prepared using the Illumina TruSeq protocol and sequenced on an Illumina Genome Analyzer II. The reads were aligned to the *Hpa* Emoy2 v8.3[9] using BWA[37] version 0.5.8. Trailing nucleotides with a quality score of less than 10 were trimmed using the -q option. In order to maximize the number of aligned reads, unaligned reads were re-aligned using Stampy[38]. SAM-tools[39] version 0.1.18 was used to generate BAM files.

Genetic variations between *Hpa* Emoy2 and each of the genome-sequenced isolates were predicted using SAMtools[39] version 0.1.18. Polymorphisms detailed in Table 1 and Supplementary Data 1 and 2 were filtered to have a minimum quality score of 10, depth of 20 and maximum depth of 500. These variant calls, including indels, were fed into SnpEff[40] version 3.2 which predicted the effect of polymorphisms to each *Hpa* Emoy2 gene in each sequenced isolate, which could then be sorted into those encoding synonymous and non-synonymous changes. To predict recognized effector candidates by association, homozygous polymorphisms were first compared across isolates using PileLine[41] version 1.2, and then selected according to the known pattern of recognition. For example, to identify *RPP4*-recognized effector candidates, genes encoding secreted proteins with polymorphisms encoding non-synonymous changes in all isolates except Emoy2 and Emwa1 were queried. Polymorphisms were visualized and confirmed in BAM files with the Integrative Genomics Viewer[42]. Finally, to generate predicted

sequences from each isolate, the *Hpa* Emoy2 v8.3 genome sequence[9] was corrected by substituting SNVs using a custom Perl script.

**Plasmid construction.** For transient gene expression in *N. benthamiana*, HaRxL103[Emoy2] (21–401 aa), HaRxL60[Emoy2] (20–202 aa), HaRxL1b[Emoy2] (27–239 aa), HaRxL71[Emoy2] (20–466 aa), HaRxLL447[Emoy2] (21–112 aa), and *HaRxL103* alleles (21–401 aa) without signal peptide sequence were cloned and assembled (using pENTR and Gateway System) into binary vector pK7WGF2 (with 35S promoter and N-terminal GFP fusion tag)[43]. The SV40 large T-antigen NLS (PKKKRKVGG)[44] or nls (PKAAAKVGG) and NES (NELALKLAGLDINK)[45] or nes (NELALK-AAGADANK) were introduced into 5′ of HaRxL103[Emoy2] (21–401 aa) and HaRxL103[Hind2] (21–401 aa) by PCR using specific oligonucleotides coding for NLS/nls and NES/nes and assembled similarly into pK7WGF2. To generate RPP4-FLAG, fragments of *RPP4* were amplified from Col-0 gDNA for Golden Gate assembly[46,47] into binary vector pICH86988 (with 35S promoter and C-terminal FLAG fusion tag). To generate RPP4-NES/nes-FLAG, NES (NELALKLAGLDINK)[45] or nes (NELALKAAGADANK) were introduced into 3′ of RPP4 by PCR using specific oligonucleotides coding for NES/nes and assembled similarly into pICH86988. To generate GFP-NLS[Emoy2] and GFP-NLS[Hind2], both forward and reverse specific oligonucleotides coding for NLS[Emoy2] (NRRKRRMFRY) and NLS[Hind2] (NRQKRRMFRY) were designed and annealed by a temperature gradient from 95 °C to 25 °C. The annealed dsDNA of *NLS[Emoy2]* and *NLS[Hind2]* were assembled similarly into binary vector pICSL86955 (with 35S promoter, N-terminal GFP fusion tag). Prediction of NLS was done by cNLS Mapper [http://nls-mapper.iab.keio.ac.jp/cgi-bin/NLS_Mapper_form.cgi]. RPS4-FLAG construct was previously reported[48].

For transient silencing in *N. benthamiana*, pHellsgate8-GUS[49] was used as a control. To generate EDS1-RNAi, sense and anti-sense sequences specific for *NbEDS1* and *PDK* intron sequence were amplified from *N. benthamiana* gDNA and pHellsgate vector, respectively, for Golden Gate assembly[46,47] into binary vector pICSL86977 (with 35S promoter) in the order; sense *NbEDS1* fragment, *PDK* intron sequence, and anti-sense *NbEDS1* fragment.

For estradiol-inducible constructs, *GFP-HaRxL103[Emoy2]* and *GFP-HaRxL103[Hind2]* were amplified from pK7WGF2-GFP-HaRxL103[Emoy2] and pK7WGF2-GFP-HaRxL103[Hind2], respectively, for Golden Gate assembly[46,47] into binary vector pICSL86933 (with 35S promoter fused to the LexA operator) containing a chimeric transcription activator XVE[50].

**Transient gene expression and plant transformation.** For transient gene expression analysis, *Agrobacterium tumefaciens* strain AGL1 was used to deliver respective transgenes in *N. benthamiana* leaves, using methods previously described[51]. For co-expression, all bacterial suspensions carrying individual constructs were adjusted to OD_{600} = 0.5 in the final mix for infiltration. Background of leaf images in Figs. 1b, 2a, 3c, and 4b and Supplementary Figs. 4b and 5c were removed by using Adobe Photoshop Elements 15. Unprocessed leaf images are provided as a Source Data file.

For plant transformation, Arabidopsis Col-0 plant and Col-0 *rpp4* mutant were transformed using the dipping method[52]. Briefly, flowering Arabidopsis plants were dipped with *A. tumefaciens* carrying a plasmid of interest, and the seeds were harvested to select the T1 transformants on selective MS media. T1 plants were checked for expression of the construct of interest by western blotting analysis. T2 seeds were sown on selective MS media, and the proportion of resistant versus susceptible plants was counted in order to identify lines with single T-DNA insertion. Transformed plants were transferred to soil and seeds collected. Two independent T3 homozygous lines were analyzed.

**RNA extraction, cDNA synthesis, and qRT-PCR.** Total RNAs were extracted using RNeasy Plant Mini Kit (Qiagen) according to the procedure of the manufacturer. Total RNAs (1 µg) were used for generating cDNAs in a 20 µl volume reaction according to Invitrogen Superscript III Reverse Transcriptase protocol. The obtained cDNAs were diluted five times, and 1 µl was used for 10 µl qPCR reaction.

qPCR was performed in 10 µl final volume using 5 µl SYBR Green mix (Toyobo), 1 µl diluted cDNAs, and primers. qPCR was run on Mx3000P qPCR System (Agilent) using the following program: (1) 95 °C, 3 min; (2) [95 °C, 30 s, then 60 °C, 30 s, then 72 °C, 30 s] × 45, (3) 95 °C, 1 min followed by a temperature gradient from 55 to 95 °C. The relative expression values were determined using the comparative cycle threshold method ($2^{-\Delta\Delta Ct}$). *Hpa* Actin, *AtEF-1α*, and *NbEF1α* were used as reference genes for *Hpa* infections, Arabidopsis and *N. benthamiana*, respectively. Primers used for qPCR are listed in Supplementary Table 2.

**Protein extraction, immunoprecipitation, and nuclear extraction.** Leaves were ground to fine powder in liquid nitrogen and thawed in extraction buffer (50 mM Tris–HCl, pH 7.5, 150 mM NaCl, 10% glycerol, 10 mM DTT, 10 mM EDTA, 1 mM NaF, 1 mM Na_2MoO_4·2H_2O, 1% IGEPAL CA-630 from Sigma-Aldrich and 1% protease inhibitor cocktail from Sigma-Aldrich). Samples were cleared by centrifugation at 16,000*g* for 15 min at 4 °C, and the supernatant was used for total protein extracts. GFP-fused and FLAG-fused proteins were detected by anti-GFP antibody (ab290; Abcam plc) in 1:8000 dilution and anti-FLAG antibody (A8592; Sigma-Aldrich) in 1:20,000 dilution, respectively.

Immunoprecipitation was performed using μMACS GFP isolation kit and μMACS DYKDDDDK isolation kit according to the manufacturer's instructions (Miltenyi Biotec).

Nuclear extraction was done by a modified method based on that described by Xu et al.[53]. Approximately 30 g of *N. benthamiana* leaves was frozen in liquid nitrogen, ground to a fine powder and homogenized in 30 ml lysis buffer (20 mM Tris–HCl, pH 7.4, 25% glycerol, 20 mM KCl, 2 mM EDTA, 2.5 mM MgCl$_2$, 250 mM sucrose, 10 mM DTT, 1 mM PMSF and protease inhibitor cocktail from Sigma-Aldrich). The homogenate was sequentially filtered through a nylon mesh. The nuclei were pelleted by centrifugation at 1500$g$ for 10 min at 4 °C, and the supernatant was taken as a cytoplasmic protein extract. The pellet was washed three times with nuclei wash buffer (20 mM Tris–HCl, pH 7.4, 25% glycerol, 2.5 mM MgCl$_2$, 0.2% Triton X-100, 10 mM DTT, 1 mM PMSF and protease inhibitor cocktail) at 4 °C. The nuclei were then resuspended in 3 ml icecold nuclei resuspension buffer (20 mM HEPES-KOH, pH 7.9, 20% glycerol, 2.5 mM MgCl$_2$, 250 mM NaCl, 0.2 mM EDTA, 0.2% Triton X-100, 10 mM DTT, 1 mM PMSF, and protease inhibitor cocktail) and then ultracentrifuged at 34,000$g$ for 15 min at 4 °C. The pellet was resuspended in the same buffer and subjected to sonication (parameters: output, 6; duty, 40) for 4 min. After centrifugation at 21,700$g$ for 30 min at 4 °C, the supernatant was taken as a nuclear protein extract. In the wash step of immunoprecipitation of cytoplasmic and nuclear protein extracts, lysis buffer and nuclei resuspension buffer were used, respectively. UGPase and Histone H3 were detected by anti-UGPase antibody (AS05 086; Agrisera) in 1:3000 dilution and anti-Histone H3 antibody (AS10 710; Agrisera) in 1:5000 dilution as markers for cytoplasmic and nuclear proteins, respectively. For the western blotting probed with antibodies against GFP and FLAG, 2 μl of cytoplasmic protein sample and 25 μl of nuclear protein sample were loaded. For the anti-UGPase and anti-Histone H3 western blotting, 5 μl of cytoplasmic protein sample and 5 μl of nuclear protein sample were loaded.

**Confocal microscopy**. For in planta subcellular localization analysis in *N. benthamiana*, cut leaf patches were mounted in water and analyzed on a Leica DM6000B/TCS SP5 confocal microscope (Leica Microsystems) with the following excitation wavelengths: GFP, 488 nm; RFP, 561 nm. In the case of Arabidopsis transformants containing Est-GFP, Est-103, and Est-103$^{Hind2}$, 3-week-old transgenic lines 24 h after spray treatment with estradiol were DAPI-stained using CySain® UV Precise P (Sysmex) by vacuum infiltration. After incubation in dark for 1 h, cut leaf patches were mounted in water and analyzed on a Leica TCS SP8 X confocal microscope (Leica Microsystems) with the following excitation wavelengths: GFP, 488 nm; DAPI, 405 nm.

**Creation of outcrossed progeny between *Hpa* Emoy2 and Cala2**. For creation of outcrossed F1 progeny, mixed inoculum of Emoy2 and Cala2 was inoculated in 10-day old seedlings of an accession compatible to both Emoy2 and Cala2. Two weeks after inoculation, the leaf tissue was frozen and dried. Oospores were left to mature for 1 month before asexual progeny were recovered (the following steps). Dried leaf tissue containing the mature oospores was ground to a fine powder using a pestle and mortar. Oospore inoculum was sprinkled onto the surface of soil in pots and seeds of a susceptible genotype were sown on top. Pots were watered and stored for 2 weeks at 4 °C to break any remaining seed dormancy. Sealed trays containing the pots were incubated in a growth cabinet at 16 °C under a 10-h photoperiod. Seedlings were inspected daily for asexual conidiosporangia from 5 days post-incubation. Individual infected seedlings bearing conidiosporangiophores were harvested and the asexual inoculum was bulked on susceptible seedlings prior to testing. Putative F1 progeny were single-spored from conidiosporangia and confirmed as hybrids by a PCR-based CAPS marker using restriction enzyme BspDI and specific primers (HaRxL103_CAPS_F and HaRxL103_CAPS_R) listed in Supplementary Table 2. The F2 population was derived from selfing the F1 and recovering progeny as described. The F2 progeny were single-spored prior to testing on CW84 and CW84:RPP4$^{Col}$. Genotyping in F2 progeny was performed by a PCR-based CAPS marker using the same restriction enzyme and primers as above.

## Data availability

The Illumina sequence data for Emoy2, Emwa1, Cala2, Emco5, Maks9, and Hind2 have been deposited in the European Nucleotide Archive (ENA) under project number PRJEB22892. The source data underlying Figs. 1b, 2a–d, 3c, d, 4b, 5a–d, and 6b and Supplementary Figs. 2a–c, 3, 4b–c, and 5a–c are provided as a Source Data file.

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

## Acknowledgements

We thank Dr. Sylvestre Marillonnet for Golden Gate vectors and Prof. Jim L. Beynon for DNAs from Emoy2/Maks9 F2 progeny. This work was supported by the Gatsby Foundation (http://www.gatsby.org.uk/); JSPS KAKENHI 15K18651 (S.A.), 17K07679 (S.A.), 17H06172 (K.S.), and 15H05959 (K.S.); RIKEN Special Postdoctoral Research Fellowship (S.A.); BBSRC BB/M003809/1 (O.J.F. and D.S.K.), BB/L011646/1 (V.C.), BB/K009176/1 (D.S.K.). We also thank Matthew Smoker and Jodie Taylor for help with Arabidopsis transformation and Takuya Okubo and Soshi Tsuchiya for their support.

## Author contributions

S.A., O.J.F., V.C., D.S.K., N.I. and S.G. conducted experiments. S.A., V.C., B.J.S., K.S. and J.D.G.J. conceived and supervised the study. S.A., K.S. and J.D.G.J. wrote the manuscript. All authors reviewed and approved the manuscript.

## Additional information

**Competing interests:** The authors declare no competing interests.

