## [Peer Review File · Nature Communications]

Reviewers' Comments:

Reviewer #1:

Remarks to the Author:

The manuscript by Asai et al identified HaRxL103 as an avr gene recognized by Arabidopsis RPP4. The authors demonstrate a mechanism that HaRxL103 natural variants evade RPP4 recognition by changing in planta subcellular localization or by lacking expression. The manuscript is well written and experiments well executed, but I do have some major concerns as outlined below.

The localization data in this manuscript are all based on transient expression in *N. benthamiana*. I strongly suggest the author check the localization of GFP-HaRxL103 in Col-0 and *rpp4*, localization of GFP-HaRxL103Hind2 in Col-0. I wonder whether the localization of GFP-HaRxL103 are similar when induced by Est in Arabidopsis or transiently expressed in *N. benthamiana*.

As is shown in Figure 5, HaRxL103 interacts with RPP4 in both cytoplasm and nucleus. This result raises a question that whether HaRxL103Hind2 can still interact with RPP4 in cytoplasm. If it can, how HaRxL103Hind2 evade recognition by RPP4 in cytoplasm? Or, in another word, cytoplasmic HaRxL103Hind2-RPP4 interaction is not essential for recognition? Meanwhile, whether NLS-HaRxL103Hind2 can still interact with RPP4 in nucleus?

A detailed analysis of RPP4-HaRxL103, RPP4-HaRxL103Hind2 interactions using BiFC or Y2H is required to improve this manuscript.

Reviewer #2:

Remarks to the Author:

Pathogen-host plant interaction is an endless topic while the evasion of recognition by host is remaining largely unknown. Identify new components in plant-microbe interaction and explore the mechanism of evading host surveillance is an important topic. In this manuscript, the authors identified a new Hpa AVR effector HaRxL103 that is recognized by Arabidopsis R gene RPP4 and they claimed that HaRxL103 evade recognition through lack of HaRxL103 expression and changing subcellular localization in host cells. The experiments are well designed and performed, which could support their conclusion. These findings add to our understanding of plant-microbe interaction. This is an interesting paper. To strengthen the manuscript, I have some comments here.

1) Major concerns:

1. The nuclear localization change caused by signal amino acid substitution is important for triggering RPP4 resistance, but the mechanism need to further test or explain.

When authors proved that HaRxL103Hind2 can evade recognition through changing subcellular localization in host cells, HaRxL103 interacts with RPP4 in cytoplasm and nucleus, but nuclear localization of HaRxL103 is required for recognition by RPP4 in planta. Does RPP4 function in cytoplasm region or in nuclear? The authors said that mutation in HaRxL103Hind2 alters in planta subcellular localization, resulting in evasion of recognition by RPP4, whether this HaRxL103Hind2 still interacts with RPP4?

2. Since authors already identify key residue, I am wondering why the authors haven't take site-directed mutagenesis of key amino acid on either HaRxL103 or HaRxL103Hind2 full length background? And it would be more physiological evidence to confirm the mutation by EST-system. The nuclear localization changed by single amino acid substitution gain virulence on PRR4 plant should be finally confirmed in Arabidopsis-Hpa disease system and scored the spores number. Only HR phenotype on benth plant is not robust to this reviewer.

2) Minor concerns:

1. It's better to show whole data of expression of HaRxL103 of Emoy2 in Figure 1c.
2. How to explain the data showed in Supplementary Figure 6b measured at different dpi? (Conidiospores were harvested and counted at 7 dpi and 5 dpi for Emoy2 and Waco9)
3. When GFP-HaRxL103 and RPP4-FLAG co-expressed in *N. benthamiana* leaves, co-immunoprecipitation was performed with each extract using MACS MicroBeads with GFP antibody but the FLAG antibody is worthwhile to test.
4. There some problems in references part, such as References 27 and 46.
5. In Fig 5, the authors state that HaRxL103 interacts with RPP4 in cytoplasm and nucleus. However, more controls are required here. The author should add HaRxL44 and other RPP genes as control.

Reviewer #3:

Remarks to the Author:

This manuscript is a well-written paper that reports an exciting novel finding in the field of plant immunity.

Essentially, plants are known to recognise pathogens by detecting directly or indirectly molecular cues and trigger an effective defence response which leads to reduced infection/enhanced resistance. In certain systems, these pathogen cues are proteins which are encoded by so-called "avirulence genes". Many of the avirulence proteins produced by the microbes are delivered to the host cells, where their main function is (counterintuitively) to enhance disease. Pathogens are therefore under selective pressures to evolve mechanisms to escape this detection. This scenario is well documented across the whole gamut of plant-microbe and plant-pest interactions.

The novelty of the submitted paper lies in the claim that in the specific case presented the pathogen escape recognition by either reducing the expression of the protein, or by altering the ultimate destination for the protein in the host cell. This is in contrast to most other well-studied cases where the avirulence proteins are (1) removed by gene deletion, inactivation, destabilisation or (2) the sequence/structures are altered to avoid recognition.

The main thrust of the message is supported by an exceptionally complex and robust set of data based on genetic and reverse genetic methodologies that are, on the whole, convincing and authoritative.

My only (broad) concern with the key message of this research is that ALL of the studies related to the localisation of avirulence proteins are based on overexpression of (tagged) proteins in a "model" plant system (*N. benthamiana*). Whilst this is understandable given the well-known limitations of working with a (so far) un-transformable obligate oomycete, there are caveats about the use of this system which need to be made explicit clearly at some (preferably several places): we might be observing here "artefacts" related to over-expression in *N. benthamiana*, rather than what is happening in the "real" *Arabidopsis*-*Hpa* system. It would be great to see at least SOME attempt at verifying the hypothesis in the native situation, perhaps using advanced proteomics approaches that should be readily accessible in the world-leading institutions of the authors. At the very least, clear disclaimers need to be made formally.

The following are minor comments /edits the authors should consider in order to improve the clarity/accuracy of the paper.

L38-39 State explicitly what the localisation is, and to which plant this refers to (i.e. the model, rather than the *Arabidopsis*-*Hpa* system).

L43 Plants... pathogens.... Offensive... defensive. This is the wrong way round. I think it should be: Plants... Pathogens.... defensive... offensive

L44 Effectors can also have other roles in addition to "attenuation of immunity". Acknowledge this, in passing, here.

L51 delete "dramatic"... this is unnecessary here. Even small changes in fitness can result in

significant evolution. C Darwin made a cogent case for that a few years back.

L65 & L68-69 I cannot quite "get" the difference between (1) and (4): can you make this clearer?

L85 All selection is "adaptive". Maybe you mean "disruptive" or "diversifying" selection as opposed to "purifying" selection?

L127-128 What happens if OTHER RPP genes are co-expressed with these effectors? Has this been tested? It would be interesting to know.

L166-L169 This is the least convincing part of the paper. Unfortunately, it is also the core experiment. There are several well-documented cases where the localisation of tagged proteins does not reflect the "true / in vivo" localisation (hence, for example, attempts at developing alternative and highly complex methodologies for the localisation of effectors in planta DOI: 10.1111/nph.14188).

L183 "less accumulation"? This data is not convicting and is in stark contrast with very rigorous approaches elsewhere in the paper to quantitative analysis.

L238 Replace "showed" with "was": is this what was intended?

L272 Substitute "disruptive" or "diversifying" instead of "positive". I know that these terms are sometimes used interchangeably, but even "purifying" selection is "positive" therefore it is best to use the alternatives.

L280-285 See comments above, with regards to necessary caveats with regards to over expression of tagged proteins in "Model" systems.

L315 and L316. Two "propose" in close proximity. Find alternative.

L351-356 Same comment as L280-285

Reviewer #4:

Remarks to the Author:

The current study is quite interesting, and represents one of only a handful of studies that have identified avirulence-R gene pairs in the Arabidopsis-downy mildew system. As such, this work is potentially transformative, as it provides a blueprint for work in other, i.e., non-model, systems. The current study clearly demonstrates that alleles within which the NES is altered result in a non-function effector, and thus, forms the basis for the recognition, or lack thereof, of this pathogen. This is really cool; indicates that pathogens can evade recognition not only by losing, or acquiring, effectors, but can modify existing effectors to redirect (on a sub-cellular level) to alter the virulence function. Very cool.

The science is solid, and while this review will not critique it (at all), the writing is dreadful. It really needs a dedicated native English speaker to edit. It was so difficult to extract the meaning, and impact, of this work. I was able to deduce some of what the authors were presenting, yet I admittedly became frustrated at the poor presentation. Below are just a few snippets of the errors – ultimately, these have led me to lose interest in evaluating this manuscript. As noted above, I am confident that this is a strong story, one with important implications, both in the approach for discovery, as well as the biological activity of these Avr-R interactions. However, the presentation does not match the science, and I therefore could not focus on a scientific critique. Here are a few examples (note that I did not make it to the end of the manuscript):

Line 91-92: In the transcriptome data by RNA sequencing. Omitted the word "generated."

Line 98: "...and their structural features after selection...." This sentence does not make sense. Odd transition within. Are you trying to say that you analyzed the (predicted) structural features, or that previous comprehensive functional screens have evaluated structural features?

Line 102: Re-write to emphasize that these effectors were identical to alleles present in Hpa Emwa1, yet were found to be polymorphic in Waco9. The mention of expressed in Waco9 needs to be re-worked. It's tagged onto the end of the sentence, and as a result, the sentence is convoluted.

Line 113: Long sentence, loses impact and is a bit convoluted: "...we confirmed it by qRT-PCR."

Line 117: Inoculated into or onto. Also, as written, it might appear that there are eds1-1 mutants that are resistant. In general, this sentence, like many others, is simply poorly constructed, lacking articles (i.e., the), commas, proper punctuation, etc.

Line 119-120: As written it sounds as if there is a subclass of NLR proteins that contain an N-terminal TIR domain within which includes RPP4. Meaning...RPP4 is found within the TIR domain. Again, poor presentation.

Line 126: Poorly constructed.

Line 135. Not sure I would use the word remarkable. Maybe substantial? Remarkable would indicate some level of "unbelievable" or "surprising."

Line 315-317: Confusing.....nonsensical.

Line 319: We suppose? Maybe hypothesize is a better choice.

Line 328: Change "...are believe to contribute...." to "...are hypothesized to contribute to virulence....":

Line 360: The nucleolus IS the site....not known as the site.

Response to the comments of **Reviewer #1**

Thank you very much for your helpful comments and critique of our manuscript (NCOMMS-17-30562). We fully revised the manuscript in line with your comments, and described the points in the text.

The manuscript by Asai et al identified HaRxL103 as an avr gene recognized by Arabidopsis RPP4. The authors demonstrate a mechanism that HaRxL103 natural variants evade RPP4 recognition by changing in planta subcellular localization or by lacking expression. The manuscript is well written and experiments well executed, but I do have some major concerns as outlined below.

The localization data in this manuscript are all based on transient expression in *N. benthamiana*. I strongly suggest the author check the localization of GFP-HaRxL103 in Col-0 and *rpp4*, localization of GFP-HaRxL103^{Hind2} in Col-0. I wonder whether the localization of GFP-HaRxL103 are similar when induced by Est in Arabidopsis or transiently expressed in *N. benthamiana*.

Answer:

As you pointed out, we checked subcellular localization of GFP-HaRxL103^{Emoy2} and GFP-HaRxL103^{Hind2} in Arabidopsis transformants. In the transformants, similar fluorescent signal patterns to ones in *N. benthamiana* transiently expressed were observed after estradiol treatment (new Figure 3e and new Supplemental Figure 7a).

We added and redrafted the sentences in Results as follows;

“In Col-0 transformants containing estradiol-inducible GFP-HaRxL103^{Emoy2} (Est-103) and GFP-HaRxL103^{Hind2} (Est-103^{Hind2}), similar fluorescent signal patterns to ones in *N. benthamiana* transiently expressed were observed (Fig. 3b, d and e).”

(L166-168)

As is shown in Figure 5, HaRxL103 interacts with RPP4 in both cytoplasm and nucleus. This result raises a question that whether HaRxL103^{Hind2} can still interact with RPP4 in cytoplasm. If it can, how HaRxL103^{Hind2} evade recognition by RPP4 in cytoplasm? Or, in another word, cytoplasmic HaRxL103^{Hind2}-RPP4 interaction is not essential for recognition? Meanwhile, whether NLS-HaRxL103^{Hind2} can still interact with RPP4 in nucleus?

Answer:

We did CoIP analysis in cytoplasmic and nuclear fractions of plant cells overexpressing GFP-HaRxL103^{Hind2} and GFP-NLS/nls-HaRxL103^{Hind2} with RPP4-FLAG (new Figure

5b, c and d). The analysis revealed that GFP-HaRxL103^{Hind2} and GFP-NLS-HaRxL103^{Hind2} interact with RPP4-FLAG in cytoplasm and nucleus, respectively. That means that the mutation in HaRxL103^{Hind2} does not affect the interaction with RPP4 and that the cytoplasmic interaction is not essential for immune responses.

We added and redrafted the sentences in Results and Discussion as follows;

“We also checked whether GFP-HaRxL103^{Hind2} and NLS/nls-fused GFP-HaRxL103^{Hind2} interact with RPP4-FLAG in cytoplasm and/or nucleus (Fig. 5b, c and d). GFP-HaRxL103^{Hind2} and GFP-nls-HaRxL103^{Hind2} were detected only in cytoplasmic fractions, whereas GFP-NLS-HaRxL103^{Hind2} was detected only in nuclear fractions. We confirmed *in planta* interaction of GFP-HaRxL103^{Hind2} and GFP-NLS/nls-HaRxL103^{Hind2} with RPP4-FLAG in each fraction, suggesting that the mutation in HaRxL103^{Hind2} does not affect the interaction with RPP4.”

(L204-210)

“In addition, co-immunoprecipitation analysis revealed the interaction of HaRxL103^{Emoy2} and HaRxL103^{Hind2} with RPP4 in cytoplasm and/or nucleus (Fig. 5).”

(L343-345)

A detailed analysis of RPP4-HaRxL103, RPP4-HaRxL103^{Hind2} interactions using BiFC or Y2H is required to improve this manuscript.

Answer:

We tried BiFC analysis by coexpressing HaRxL103 with RPP4, but we failed to observe fluorescent signals. In addition, we could observe no fluorescent signal in plant cells expressing RPP4-GFP. It might be due to turnover of RPP4-GFP.

Again, thank you very much for your helpful critical comments.

Response to the comments of **Reviewer #2**

Thank you very much for your helpful comments and critique of our manuscript (NCOMMS-17-30562). We fully revised the manuscript in line with your comments, and described the points in the text.

Pathogen-host plant interaction is an endless topic while the evasion of recognition by host is remaining largely unknown. Identify new components in plant-microbe interaction and explore the mechanism of evading host surveillance is an important topic. In this manuscript, the authors identified a new Hpa AVR effector HaRxL103 that is recognized by Arabidopsis R gene RPP4 and they claimed that HaRxL103 evade recognition through lack of HaRxL103 expression and changing subcellular localization in host cells. The experiments are well designed and performed, which could support their conclusion. These findings add to our understanding of plant-microbe interaction. This is an interesting paper. To strengthen the manuscript, I have some comments here.

1) Major concerns:

1. The nuclear localization change caused by signal amino acid substitution is important for triggering RPP4 resistance, but the mechanism need to further test or explain. When authors proved that HaRxL103^{Hind2} can evade recognition through changing subcellular localization in host cells, HaRxL103 interacts with RPP4 in cytoplasm and nucleus, but nuclear localization of HaRxL103 is required for recognition by RPP4 in planta. Does RPP4 function in cytoplasm region or in nuclear? The authors said that mutation in HaRxL103^{Hind2} alters in planta subcellular localization, resulting in evasion of recognition by RPP4, whether this HaRxL103^{Hind2} still interacts with RPP4?

Answer:

We did CoIP analysis in cytoplasmic and nuclear fractions of plant cells overexpressing GFP-HaRxL103^{Hind2} and GFP-NLS/nls-HaRxL103^{Hind2} with RPP4-FLAG (new Figure 5b, c and d). The analysis revealed that GFP-HaRxL103^{Hind2} and GFP-NLS-HaRxL103^{Hind2} interact with RPP4-FLAG in cytoplasm and nucleus, respectively. That means that the mutation in HaRxL103^{Hind2} does not affect the interaction with RPP4 and that the cytoplasmic interaction is not essential for immune responses.

We added and redrafted the sentences in Results and Discussion as follows;

“We also checked whether GFP-HaRxL103^{Hind2} and NLS/nls-fused GFP-HaRxL103^{Hind2} interact with RPP4-FLAG in cytoplasm and/or nucleus (Fig. 5b, c and d). GFP-HaRxL103^{Hind2} and GFP-nls-HaRxL103^{Hind2} were detected only in cytoplasmic

fractions, whereas GFP-NLS-HaRxL103^{Hind2} was detected only in nuclear fractions. We confirmed *in planta* interaction of GFP-HaRxL103^{Hind2} and GFP-NLS/nls-HaRxL103^{Hind2} with RPP4-FLAG in each fraction, suggesting that the mutation in HaRxL103^{Hind2} does not affect the interaction with RPP4.”

(L204-210)

“In addition, co-immunoprecipitation analysis revealed the interaction of HaRxL103^{Emoy2} and HaRxL103^{Hind2} with RPP4 in cytoplasm and/or nucleus (Fig. 5).”

(L343-345)

To give an answer to your question whether RPP4 functions in cytoplasm and/or in nucleus, NES/nls-fused RPP4-FLAG in N-terminus or C-terminus were constructed. As our hypothesis was that nucleus-localized RPP4 activates immune responses in presence of HaRxL103, we expected that NES-RPP4 does not function. However, NES-fused RPP4-FLAG induced HR cell death at the same levels as RPP4-FLAG when co-expressed with GFP-HaRxL103 in *N. benthamiana*. We cannot draw a conclusion from this result because we have not checked if NES-fused RPP4-FLAG is excluded from nucleus. Cytoplasmic RPP4 might be carried to nucleus by HaRxL103. Further experiments are required to conclude it. We believe that this is not essential for the conclusions in this manuscript.

2. Since authors already identify key residue, I am wondering why the authors haven't take site-directed mutagenesis of key amino acid on either HaRxL103 or HaRxL103^{Hind2} full length background? And it would be more physiological evidence to confirm the mutation by EST-system. The nuclear localization changed by single amino acid substitution gain virulence on PRR4 plant should be finally confirmed in Arabidopsis-Hpa disease system and scored the spores number. Only HR phenotype on benth plant is not robust to this reviewer.

Answer:

As shown in Figure 3a and Supplemental Figure 5, *HaRxL103^{Hind2}* contains only one non-synonymous SNV compared to *HaRxL103^{Emoy2}*. The mutation affects its subcellular localization in plant cells. As you pointed out, we measured *Hpa* growth in Col-0 transformants containing estradiol-inducible GFP-HaRxL103^{Emoy2} (Est-103) and GFP-HaRxL103^{Hind2} (Est-103^{Hind2}). Consistent with HR cell death phenotypes in *N. benthamiana* (Figure 3c) and *PR1* expression in Col-0 Est-103 and Est-103^{Hind2} (Figure 3f), *Hpa* sporulated on Col-0 Est-103^{Hind2}, but not Col-0 Est-103, pretreated with estradiol (new Figure 3g).

We added and redrafted the sentences in Results as follows;

“Consistent with HR phenotypes in *N. benthamiana*, Col-0 Est-103^{Hind2} showed less induction of *PR1* than Col-0 Est-103 after treatment with estradiol (Fig. 3f). *Hpa* sporulated on Col-0 Est-103^{Hind2}, but not Col-0 Est-103, pretreated with estradiol (Fig. 3g).”

(L169-171)

2) Minor concerns:

1. It's better to show whole data of expression of HaRxL103 of Emoy2 in Figure 1c.

Answer:

In the *Hpa* Emoy2 and Col-0 interaction (an incompatible interaction), there was not enough RNA derived from Emoy2 to be detected in the infections after 1 dpi as described in our previous work (PLOS Pathogens 10: e1004443, 2014).

2. How to explain the data showed in Supplementary Figure 6b measured at different dpi? (Conidiospores were harvested and counted at 7 dpi and 5 dpi for Emoy2 and Waco9)

Answer:

Hpa Waco9 is higher virulent isolate than *Hpa* Emoy2. Therefore, 7 dpi and 5 dpi are the best timings to measure conidiospores formed on infections of *Hpa* Emoy2 and Waco9, respectively.

3. When GFP-HaRxL103 and RPP4-FLAG co-expressed in *N. benthamiana* leaves, co-immunoprecipitation was performed with each extract using MACS MicroBeads with GFP antibody but the FLAG antibody is worthwhile to test.

Answer:

As you suggested, we performed Co-IP analysis using MACS MicroBeads with FLAG antibody. Corresponding to the result using MACS MicroBeads with GFP antibody, we confirmed *in planta* interaction between GFP-HaRxL103 and RPP4-FLAG (new Supplemental Figure 3).

4. There some problems in references part, such as References 27 and 46.

Answer:

We corrected it.

5. In Fig 5, the authors state that HaRxL103 interacts with RPP4 in cytoplasm and nucleus. However, more controls are required here. The author should add HaRxL44 and other RPP genes as control.

Answer:

As you suggested, we checked if HaRxL103 interacts with RPS4, a TIR-NLR related to recognition of AvrRps4 and PopP2 (Plant Journal 60: 218–226, 2009). Co-IP analysis showed that GFP-HaRxL103 does not interact with RPS4-FLAG (new Supplemental Figure 3). We confirmed no *in planta* interaction between GFP and RPP4-FLAG (Figure 2c and new Supplemental Figure 3). We believe that GFP is a control for GFP-HaRxL103.

Again, thank you very much for your helpful critical comments.

Response to the comments of **Reviewer #3**

Thank you very much for your helpful comments and critique of our manuscript (NCOMMS-17-30562). We fully revised the manuscript in line with your comments, and described the points in the text.

This manuscript is a well-written paper that reports an exciting novel finding in the field of plant immunity.

Essentially, plants are known to recognise pathogens by detecting directly or indirectly molecular cues and trigger an effective defence response which leads to reduced infection/enhanced resistance. In certain systems, these pathogen cues are proteins which are encoded by so-called “avirulence genes”. Many of the avirulence proteins produced by the microbes are delivered to the host cells, where their main function is (counterintuitively) to enhance disease. Pathogens are therefore under selective pressures to evolve mechanisms to escape this detection. This scenario is well documented across the whole gamut of plant-microbe and plant-pest interactions.

The novelty of the submitted paper lies in the claim that in the specific case presented the pathogen escape recognition by either reducing the expression of the protein, or by altering the ultimate destination for the protein in the host cell. This is in contrast to most other well-studied cases where the avirulence proteins are (1) removed by gene deletion, inactivation, destabilisation or (2) the sequence/structures are altered to avoid recognition.

The main thrust of the message is supported by an exceptionally complex and robust set of data based on genetic and reverse genetic methodologies that are, on the whole, convincing and authoritative.

My only (broad) concern with the key message of this research is that ALL of the studies related to the localisation of avirulence proteins are based on overexpression of (tagged) proteins in a “model” plant system (*N. benthamiana*). Whilst this is understandable given the well-known limitations of working with a (so far) un-transformable obligate oomycete, there are caveats about the use of this system which need to be made explicit clearly at some (preferably several places): we might be observing here “artefacts” related to over-expression in *N. benthamiana*, rather than what is happening in the “real” *Arabidopsis*-*Hpa* system. It would be great to see at least SOME attempt at verifying the hypothesis in the native situation, perhaps using

advanced proteomics approaches that should be readily accessible in the world-leading institutions of the authors. At the very least, clear disclaimers need to be made formally.

Answer:

We checked subcellular localization of GFP-HaRxL103^{Emoy2}, GFP-HaRxL103^{Hind2} and GFP-NLS/nls-HaRxL103^{Hind2} in Arabidopsis transformants. In the transformants, similar fluorescent signal patterns to ones in *N. benthamiana* transiently expressed were observed after estradiol treatment (new Figure 3e and new Supplemental Figure 7a).

We added and redrafted the sentences in Results as follows;

“In Col-0 transformants containing estradiol-inducible GFP-HaRxL103^{Emoy2} (Est-103) and GFP-HaRxL103^{Hind2} (Est-103^{Hind2}), similar fluorescent signal patterns to ones in *N. benthamiana* transiently expressed were observed (Fig. 3b, d and e).”

(L166-168)

In addition, we measured *Hpa* growth in Col-0 transformants containing estradiol-inducible GFP-HaRxL103^{Emoy2} (Est-103) and GFP-HaRxL103^{Hind2} (Est-103^{Hind2}). Consistent with HR cell death phenotypes in *N. benthamiana* (Figure 3c) and *PR1* expression in Col-0 Est-103 and Est-103^{Hind2} (Figure 3f), *Hpa* sporulated on Col-0 Est-103^{Hind2}, but not Col-0 Est-103, pretreated with estradiol (new Figure 3g).

We added and redrafted the sentences in Results as follows;

“Consistent with HR phenotypes in *N. benthamiana*, Col-0 Est-103^{Hind2} showed less induction of *PR1* than Col-0 Est-103 after treatment with estradiol (Fig. 3f). *Hpa* sporulated on Col-0 Est-103^{Hind2}, but not Col-0 Est-103, pretreated with estradiol (Fig. 3g).”

(L169-171)

In infections, native proteins of pathogen effectors and plant NLRs are a too small amount for detection. It would be, therefore, difficult to verify our hypothesis in the native situation. We believe that these additional experiments strengthen the conclusions in this manuscript.

Also, as you suggested, we added a caveat in Discussion as follows;

“In this study, GFP-tagged proteins were ectopically overexpressed to check those subcellular localizations. Although we cannot rule out the possibility that the localization of GFP-tagged HaRxL103 does not reflect the real localization of native HaRxL103 proteins during *Hpa* infection, *in planta* subcellular localization of HaRxL103 correlated with *RPP4*-mediated immunity in the conditions tested.”

(L281-286)

The following are minor comments /edits the authors should consider in order to improve the clarity/accuracy of the paper.

L38-39 State explicitly what the localisation is, and to which plant this refers to (i.e. the model, rather than the Arabidopsis-Hpa system).

Answer:

We redrafted the sentence as follows;

“We report here the identification of the Emoy2 *AVR* effector gene recognized by *RPP4* and show resistance-breaking isolates of *Hpa* on *RPP4*-containing Arabidopsis carry the alleles that either are not expressed, or show cytoplasmic instead of nuclear subcellular localization.”

(L34-37)

L43 Plants... pathogens.... Offensive... defensive. This is the wrong way round. I think it should be: Plants... Pathogens.... defensive... offensive

Answer:

We redrafted the sentence as follows;

“Plants and pathogens have co-evolved in a defensive and offensive battle for survival.”

(L40)

L44 Effectors can also have other roles in addition to “attenuation of immunity”. Acknowledge this, in passing, here.

Answer:

We redrafted the sentence as follows;

“Pathogens promote infection success by secreting effector proteins that modulate a variety of plant cellular functions, thus rendering hosts more susceptible.”

(L41-42)

L51 delete “dramatic”... this is unnecessary here. Even small changes in fitness can result in significant evolution. C Darwin made a cogent case for that a few years back.

Answer:

We redrafted the sentence as follows;

“Allelic variation, including loss-of-function mutations, in an *AVR* gene can enable a pathogen race to evade recognition and cause disease on plants that carry the cognate *R* gene.”

(L48-50)

L65 & L68-69 I cannot quite “get” the difference between (1) and (4): can you make this clearer?

Answer:

To make clear, we redrafted the sentence as follows;

“(1) proteins with a signal peptide and canonical RxLR motif, like ATR1, ATR13 and ATR39 (HaRxLs)^{4,6,7}, reported by Baxter et al (2010)⁹ (2) RxLR-like proteins with at least one non-canonical feature, like ATR5 (HaRxLLs)⁵, (3) putative Crinkler-like proteins with RxLR motif (HaRxLCRN)¹¹, (4) homologous proteins based on amino acid sequence similarity over the 5’ region including a signal peptide and RxLR motif (e.g., HaRxL1b, HaRxLL2b and HaRxLCRN3b).”

(L60-66)

L85 All selection is “adaptive”. Maybe you mean “disruptive” or “diversifying” selection as opposed to “purifying” selection?

Answer:

We changed the word from “adaptive” to “diversifying”.

(L82)

L127-128 What happens if OTHER RPP genes are co-expressed with these effectors? Has this been tested? It would be interesting to know.

Answer:

We checked if HaRxL103 interacts with RPS4, a TIR-NLR related to recognition of AvrRps4 and PopP2 (60, 218–226 Plant J. 2009). Co-immunoprecipitation analysis showed that GFP-HaRxL103 does not interact with RPS4-FLAG (new Supplemental Figure 3).

L166-L169 This is the least convincing part of the paper. Unfortunately, it is also the core experiment. There are several well-documented cases where the localisation of tagged proteins does not reflect the “true / in vivo” localisation (hence, for example, attempts at developing alternative and highly complex methodologies for the localisation of effectors in planta DOI: 10.1111/nph.14188).

Answer:

In the method you suggested (DOI: 10.1111/nph.14188), host plants stably expressing the bacterial biotin ligase *BirA* and pathogens containing effector proteins fused to a short peptide (Avitag) biotinylated by BirA are necessary. This approach is difficult due

to the difficulty of transformation in obligate oomycete *Hpa*. As described above, we confirmed similar subcellular localization of GFP-HaRxL103^{Emoy2} and GFP-HaRxL103^{Hind2} and activation of immune responses in Arabidopsis transformants to ones observed in *N. benthamiana* transiently expressed (new Figure 3e, g and new Supplemental Figure 7a). We also added a caveat in Discussion. (L281-286)

L183 “less accumulation”? This data is not convicting and is in stark contrast with very rigorous approaches elsewhere in the paper to quantitative analysis.

Answer:

As shown in new Figure 5b, GFP-HaRxL103^{Hind2} was detected in only cytoplasmic fraction. It is likely to mean that accumulation levels of HaRxL103^{Hind2} in nucleus (and nucleolus) are less than ones of HaRxL103^{Emoy2}. However, we did not perform a quantitative analysis using microscopic images. To avoid misunderstanding, we changed the word “less” throughout the manuscript as follows;

“only a weak fluorescent signal” (L161)

“little accumulation” (L172)

“little accumulation” (L180)

“little nucleolar localization” (L192)

“reduced accumulation” (L341)

L238 Replace “showed” with “was”: is this what was intended?

Answer:

We changed the word from “showed” to “was”.

(L236)

L272 Substitute “disruptive” or “diversifying” instead of “positive”. I know that these terms are sometimes used interchangeably, but even “purifying” selection is “positive” therefore it is best to use the alternatives.

Answer:

We changed the word from “positive” to “diversifying”.

(L270)

L280-285 See comments above, with regards to necessary caveats with regards to over expression of tagged proteins in “Model” systems.

Answer:

As described above, we added a caveat in Discussion. (L281-286)

L315 and L316. Two “propose” in close proximity. Find alternative.

Answer:

As you suggested, we changed the word from “propose” to “suggest”.

We divided the sentence into two sentences as follows;

“We propose referring to *HaRxL103^{Emoy2}* as *AvrRPP4*, but not *ATR4*. We suggest *ATR4* should be reserved for the locus at which genetic variation is found that regulates *HaRxL103* expression.”

(L313-315)

L351-356 Same comment as L280-285

Answer:

As described above, we added a caveat in Discussion. (L281-286)

Again, thank you very much for your helpful critical comments.

Response to the comments of **Reviewer #4**

Thank you very much for your helpful comments and critique of our manuscript (NCOMMS-17-30562). We fully revised the manuscript in line with your comments, and described the points in the text.

The current study is quite interesting, and represents one of only a handful of studies that have identified avirulence-R gene pairs in the Arabidopsis-downy mildew system. As such, this work is potentially transformative, as it provides a blueprint for work in other, i.e., non-model, systems. The current study clearly demonstrates that alleles within which the NES is altered result in a non-function effector, and thus, forms the basis for the recognition, or lack thereof, of this pathogen. This is really cool; indicates that pathogens can evade recognition not only by losing, or acquiring, effectors, but can modify existing effectors to redirect (on a sub-cellular level) to alter the virulence function. Very cool.

The science is solid, and while this review will not critique it (at all), the writing is dreadful. It really needs a dedicated native English speaker to edit. It was so difficult to extract the meaning, and impact, of this work. I was able to deduce some of what the authors were presenting, yet I admittedly became frustrated at the poor presentation. Below are just a few snippets of the errors – ultimately, these have led me to lose interest in evaluating this manuscript. As noted above, I am confident that this is a strong story, one with important implications, both in the approach for discovery, as well as the biological activity of these Avr-R interactions. However, the presentation does not match the science, and I therefore could not focus on a scientific critique. Here are a few example (note that I did not make it to the end of the manuscript):

Answer:

We thoroughly rewrote the manuscript. Words and sentences other than ones you pointed out (in the followings) were also modified. We hope that the current manuscript is improved.

Line 91-92: In the transcriptome data by RNA sequencing. Omitted the word “generated.”

Answer:

We redrafted the sentence as follows;

“In *Hpa* Emoy2-infected Arabidopsis Col-0 (an incompatible interaction), transcripts from *Hpa* clearly decreased from 1 day post-inoculation (dpi), consistent with *Hpa* Emoy2 growth being arrested upon recognition by *RPP4*.”

(L87-90)

Line 98: "...and their structural features after selection..." This sentence does not make sense. Odd transition within. Are you trying to say that you analyzed the (predicted) structural features, or that previous comprehensive functional screens have evaluated structural features?

Answer:

We meant that we analyzed the predicted structural features.

We redrafted the sentences in Results as follows;

"We also examined the genomes of seven sequenced *Hpa* isolates and analyzed transcriptome data of *Hpa* Waco9. These analyses revealed five candidate *Hpa* effectors^{15,16} (Fig. 1a and Supplementary Table 3)"

(L91-93)

Line 102: Re-write to emphasize that these effectors were identical to alleles present in *Hpa* Emwa1, yet were found to be polymorphic in Waco9. The mention of expressed in Waco9 needs to be re-worked. It's tagged onto the end of the sentence, and as a result, the sentence is convoluted.

Answer:

We redrafted the sentence as follows;

"HaRxL60 and HaRxL1b were identical to alleles present in *Hpa* Emwa1, yet were found to be polymorphic in *Hpa* Waco9."

(L95-96)

Line 113: Long sentence, loses impact and is a bit convoluted: "...we confirmed it by qRT-PCR."

Answer:

We redrafted the sentence as follows;

"We confirmed that *HaRxL103* is expressed at 1 dpi in *Hpa* Emoy2, but not in *Hpa* Waco9, during infection on *Arabidopsis* Col-0 (Fig. 1c)."

(L104-106)

Line 117: Inoculated into or onto. Also, as written, it might appear that there are eds1-1 mutants that are resistant. In general, this sentence, like many others, is simply poorly constructed, lacking articles (i.e., the), commas, proper punctuation, etc.

Answer:

We redrafted the sentence as follows;

“To evaluate the expression patterns of *HaRxL103* in a compatible interaction, we inoculated *Arabidopsis enhanced disease susceptibility 1* mutant *Ws eds1-1* with *Hpa Emoy2* and *Waco9*. *Ws eds1-1* is susceptible to both *Hpa Emoy2* and *Waco9*¹⁷.”

(L106-109)

Line 119-120: As written it sounds as if there is a subclass of NLR proteins that contain an N-terminal TIR domain within which includes RPP4. Meaning....RPP4 is found within the TIR domain. Again, poor presentation.

Answer:

We redrafted the sentence as follows;

“*RPP4* encodes an N-terminal TIR domain-containing NLR (TIR-NLR). As *EDS1* is required for the function of TIR-NLR proteins¹⁸, we tested if *EDS1* is required for *RPP4* function.”

(L111-112)

Line 126: Poorly constructed.

Answer:

We redrafted the sentence as follows;

“HR cell death induced by *GFP-HaRxL103* and *RPP4-FLAG* co-expression was observed in a leaf area overexpressing hairpin RNA targeted against *GUS* (*GUS-RNAi*) as a control, whereas the HR cell death was compromised in *NbEDS1*-silenced leaf area (Fig. 2a and b).”

(L116-119)

Line 135. Not sure I would use the word remarkable. Maybe substantial? Remarkable would indicate some level of “unbelievable” or “surprising.”

Answer:

We changed the words from “a remarkable reduction in *PR1* expression” to “lower *PR1* expression”.

(L127)

Line 315-317: Confusing.....nonsensical.

Answer:

To avoid confusion, we removed the following sentence.

“We did not have access to an Emoy2/Hind2 progeny in which linkage to the HaRxL103 locus might have been detected.”

(L313)

Line 319: We suppose? Maybe hypothesize is a better choice.

Answer:

We redrafted the sentence as follows;

“The flanking sequence of the *HaRxL103*-coding region are identical in *Hpa* virulent isolate Waco9 and avirulent isolate Emoy2, so we propose that allelic variation between virulent and avirulent isolates is found in gene(s) involved in epigenetic and/or transcriptional regulation of *HaRxL103*.”

(L315-318)

Line 328: Change “....are believe to contribute....” to “....are hypothesized to contribute to virulence....”:

Answer:

We redrafted the sentence as follows;

“Effector genes that trigger ETI on their host plants are likely to be rapidly lost unless they contribute to virulence on susceptible host plants.”

(L324-325)

Line 360: The nucleolus IS the site....not known as the site.

Answer:

We redrafted the sentence as follows;

“The nucleolus is the site for ribosomal RNA synthesis and ribosome assembly.”

(L350-351)

Again, thank you very much for your helpful critical comments.

Reviewers' Comments:

Reviewer #1:

Remarks to the Author:

In this revised manuscript, the authors almost answered all my questions. The author mentioned they failed BiFC experiment, I wonder whether the author test all the BiFC combinations? Whether the author tested fluorescence signal for both N and C terminal GFP tagged RPP4? I hope the authors can clearly describe those details in their manuscript.

Reviewer #2:

Remarks to the Author:

In the revised manuscript, authors already made some changes. I think most of the concerns are addressed and are rationally explained. Thank you.

My major concern on the mechanism why HaRxL103 evades RPP4 surveillance still exists. HaRxL103Hind2 still interact with RPP4 in cytoplasmic region, NES-RPP4 trigger defense as the same level as RPP4. If all these stand, how HaRxL103Hind2 evade RPP4 recognition? Otherwise, other mechanisms may be involved?

1. Fig5 is important to support authors' conclusion and I think it is good data, but this key experiment should be verified independently through at least one other method, co-localization etc?

2. We strongly suggest authors to detect NES-RPP4 localization! RPP4 cytoplasmic localization/interaction with HaRxL103 is not essential for defense should be experimentally validated. The resistance/HR in different conditions should be checked.

3. The fluorescence intensity profile in the Fig 4 should make bigger so people could read clearly.

4. Some species names should be in italic in the reference part.

5. Should add scale bars in RFP and Merge part of Fig. 4a and Fig. 4d.

Reviewer #3:

Remarks to the Author:

I am happy with the rebuttals and the changes made

Response to the comments of **Reviewer #1**

Thank you very much for your helpful comments and critique of our manuscript (NCOMMS-17-30562). We fully revised the manuscript in line with your comments, and described the points in the text.

In this revised manuscript, the authors almost answered all my questions. The author mentioned they failed BiFC experiment, I wonder whether the author test all the BiFC combinations? Whether the author tested fluorescence signal for both N and C terminal GFP tagged RPP4? I hope the authors can clearly describe those details in their manuscript.

Answer:

As you suggested, we added the description in Results as follows;

“To evaluate whether nuclear localization of RPP4 is essential for recognition of HaRxL103^{Emoy2}, RPP4 fused to NES or nes (RPP4-NES and RPP4-nes, respectively) were constructed. As we could observe no fluorescent signals in plant cells expressing RPP4-GFP, fractionation of cytoplasmic and nuclear proteins was done to check *in planta* subcellular localization of RPP4-NES/nes.”

(L147-152)

“To test this hypothesis, we performed bimolecular fluorescence complementation analysis by co-expressing nVenus/cCFP-HaRxL103^{Emoy2} with RPP4-nVenus/cCFP in all combinations, but we failed to observe fluorescent signals. Therefore, co-immunoprecipitation in a nuclear fraction was performed.”

(L213-216)

Again, thank you very much for your helpful critical comments.

Response to the comments of **Reviewer #2**

Thank you very much for your helpful comments and critique of our manuscript (NCOMMS-17-30562). We fully revised the manuscript in line with your comments, and described the points in the text.

In the revised manuscript, authors already made some changes. I think most of the concerns are addressed and are rationally explained. Thank you.

My major concern on the mechanism why HaRxL103 evades RPP4 surveillance still exists.

HaRxL103Hind2 still interact with RPP4 in cytoplasmic region, NES-RPP4 trigger defense as the same level as RPP4. If all these stand, how HaRxL103Hind2 evade RPP4 recognition? Otherwise, other mechanisms may be involved?

1. Fig5 is important to support authors' conclusion and I think it is good data, but this key experiment should be verified independently through at least one other method, co-localization etc?

2. We strongly suggest authors to detect NES-RPP4 localization! RPP4 cytoplasmic localization/interaction with HaRxL103 is not essential for defense should be experimentally validated. The resistance/HR in different conditions should be checked.

Answer:

We gave the following comment in the 2nd submission;

“To give an answer to your question whether RPP4 functions in cytoplasm and/or in nucleus, NES/nes-fused RPP4-FLAG in N-terminus or C-terminus were constructed. As our hypothesis was that nucleus-localized RPP4 activates immune responses in presence of HaRxL103, we expected that NES-RPP4 does not function. However, NES-fused RPP4-FLAG induced HR cell death at the same levels as RPP4-FLAG when co-expressed with GFP-HaRxL103 in *N. benthamiana*.”

In that stage, we tested HR phenotypes on the same condition as the others (*Agrobacterium* suspensions were adjusted to OD₆₀₀ = 0.5 in the final mix for infiltration). This time, we checked the phenotypes using different concentrations (OD₆₀₀ = 0.1, 0.25 or 0.5) of *Agrobacterium* containing RPP4-NES/nes-FLAG. We found that HR cell death mediated by RPP4-NES-FLAG, but not RPP4-nes-FLAG, was compromised when co-expressed with GFP-HaRxL103^{Emoy2} compared to that by RPP4-FLAG (Supplementary Figure 5c).

As you suggested, we confirmed subcellular localization of RPP4-NES/nes-FLAG by fractionation of cytoplasmic and nuclear proteins. RPP4-FLAG and RPP4-nes-FLAG

were detected in both cytoplasmic and nuclear fractions, whereas RPP4-NES-FLAG was detected in a cytoplasmic fraction (Supplementary Figure 5a)

We tried BiFC analysis by co-expressing HaRxL103^{Emoy2} with RPP4, but we failed to observe fluorescent signals. In addition, we could observe no fluorescent signal in plant cells expressing RPP4-GFP. Thus, we could not check whether HaRxL103^{Emoy2} co-localizes to nucleus with RPP4 by using a fluorescent microscopy.

We added the sentences in Results as follows;

“Like HaRxL103^{Emoy2}, RPP4 is localized to cytoplasm and nucleus¹⁹. To evaluate whether nuclear localization of RPP4 is essential for recognition of HaRxL103^{Emoy2}, RPP4 fused to NES or nes (RPP4-NES and RPP4-nes, respectively) were constructed. As we could observe no fluorescent signals in plant cells expressing RPP4-GFP, fractionation of cytoplasmic and nuclear proteins was done to check *in planta* subcellular localization of RPP4-NES/nis. RPP4-FLAG and RPP4-nis-FLAG were detected in both cytoplasmic and nuclear fractions, whereas RPP4-NES-FLAG was detected in a cytoplasmic fraction (Supplementary Fig. 5a). RPP4-nis-FLAG induced HR cell death at the same levels as RPP4-FLAG, but HR cell death mediated by RPP4-NES-FLAG was compromised when co-expressed with GFP-HaRxL103^{Emoy2} in *N. benthamiana* leaves (Supplementary Fig. 5b and c). These results suggest that nuclear localization of RPP4 is important for recognition of HaRxL103^{Emoy2}.”

(L147-158)

“To test this hypothesis, we performed bimolecular fluorescence complementation analysis by co-expressing nVenus/cCFP-HaRxL103^{Emoy2} with RPP4-nVenus/cCFP in all combinations, but we failed to observe fluorescent signals. Therefore, co-immunoprecipitation in a nuclear fraction was performed.”

(L213-216)

3.The fluorescence intensity profile in the Fig 4 should make bigger so people could read clearly.

Answer:

As you suggested, we made the fluorescence intensity profile in the Fig 4 bigger.

4.Some species names should be in italic in the reference part.

Answer:

We checked titles in References carefully. We think that titles are the same as ones of publications.

5. Should add scale bars in RFP and Merge part of Fig. 4a and Fig. 4d.

Answer:

As you suggested, scale bars were added in Figure 4a and 4d.

Again, thank you very much for your helpful critical comments.

Reviewers' Comments:

Reviewer #2:

Remarks to the Author:

Now I am satisfying with the response from authors. I appreciate the novelty of the research and also understand the technical difficulties. Congratulations to all the authors.

I'd suggest authors to add a bit of discussions on some recent publications with similar topics.

(1)Variation in the AvrSr35 gene determines Sr35 resistance against wheat stem rust race Ug99.
Salcedo A et al.,Science. 2017

(2)Natural allelic variations provide insights into host adaptation of Phytophthora avirulence effector PsAvr3c.

Huang J et al., New Phytol. 2018

I may miss some other publications, but a little bit expansions in discussion will be helpful to gain insight into plant-pathogen co-evolution.

Response to the comments of **Reviewer #2**

Thank you very much for your helpful comments and critique of our manuscript (NCOMMS-17-30562). We fully revised the manuscript in line with your comments, and described the points in the text.

Now I am satisfying with the response from authors. I appreciate the novelty of the research and also understand the technical difficulties. Congratulations to all the authors. I'd suggest authors to add a bit of discussions on some recent publications with similar topics.

(1) Variation in the *AvrSr35* gene determines *Sr35* resistance against wheat stem rust race Ug99. Salcedo A et al., *Science*. 2017

(2) Natural allelic variations provide insights into host adaptation of *Phytophthora* avirulence effector *PsAvr3c*. Huang J et al., *New Phytol*. 2018

I may miss some other publications, but a little bit expansions in discussion will be helpful to gain insight into plant-pathogen co-evolution.

Answer:

We appreciate your suggestion.

We added the suggested publications in Discussion as follows;

“Virulent isolates of wheat stem rust break resistance conferred by the wheat *Sr35* resistance gene through loss of *AvrSr35* by the insertion of a mobile element²⁴. *ATR1* and *ATR13* are extremely polymorphic and this allelic diversity enables evasion of recognition by specific alleles of their corresponding *R* genes, *RPP1* and *RPP13*^{4,6,25,26}. In *Phytophthora sojae*, a key amino acid mutation in *PsAvr3c* impairs a physical association with the host protein GmSKRPs involved in *Rps3c*-mediated soybean immunity, resulting in evasion of recognition by *Rps3c*^{27,28}.”

(lines 288 to 294)

Again, thank you very much for your helpful critical comments.